# A Pairwise Pseudo-likelihood Approach for Matrix Completion with Informative Missingness

**Jiangyuan Li**[*][†]
Department of Statistics
Texas A&M University
College Station, TX 77843
jiangyuanli@tamu.edu

**Jiayi Wang**[*]
Department of Mathematical Sciences
University of Texas at Dallas
Richardson, TX 75080
jiayi.wang2@utdallas.edu

**Raymond K. W. Wong**
Department of Statistics
Texas A&M University
College Station, TX 77843
raywong@tamu.edu

**Kwun Chuen Gary Chan**
Department of Biostatistics
University of Washington
Seattle, WA 98195
kcgchan@uw.edu

## Abstract

While several recent matrix completion methods are developed to deal with non-uniform observation probabilities across matrix entries, very few allow the missingness to depend on the mostly unobserved matrix measurements, which is generally ill-posed. We aim to tackle a subclass of these ill-posed settings, characterized by a flexible separable observation probability assumption that can depend on the matrix measurements. We propose a regularized pairwise pseudo-likelihood approach for matrix completion and prove that the proposed estimator can asymptotically recover the low-rank parameter matrix up to an identifiable equivalence class of a constant shift and scaling, at a near-optimal asymptotic convergence rate of the standard well-posed (non-informative missing) setting, while effectively mitigating the impact of informative missingness. The efficacy of our method is validated via numerical experiments, positioning it as a robust tool for matrix completion to mitigate data bias.

## 1   Introduction

The goal of matrix completion is to recover a target matrix from its noisy and incomplete measurements. It is a modern high-dimensional missing data problem. Despite various significant advances made in the last two decades [e.g., 7, 17, 20, 18], many works on matrix completion still focus on the missing-at-random mechanism. Although such an assumption is doubtful in many real-life applications, there are very few options available for more general missing data mechanism, especially those with theoretical guarantees. This work aims to provide a principled and theoretically well-supported alternative method for missing-not-at-random settings, where missingness could depend on the measurements that are mostly unobserved.

A usual assumption to allow for succeeding matrix completion is that the unknown matrix is low-rank or approximately low-rank. The noiseless setting has been studied in [7] using nuclear norm minimization. The vast majority of existing theories on matrix completion assume that entries are revealed with a constant probability with respect to both entry location and measurement value.

---

[*]Equal contribution
[†]Now at Google

38th Conference on Neural Information Processing Systems (NeurIPS 2024).

Recent approaches to handling entries revealed with nonuniform probabilities, depending on the entry location, have shown the strength to improve matrix completion with solid theoretical guarantees [e.g., 31, 11, 28, 25, 23, 35]. These works aim to mitigate the effects due to the non-uniformity in observation probabilities [30]. When additional row/column attributes are available, it is also possible to use this additional information for handling the non-uniform missing [e.g., 24]. Although the non-uniform missing mechanism is quite flexible, it is fundamentally different from the missing-not-at-random mechanism. The key difference is whether the missing probability depends on the possibly unobserved measurement, which we will highlight below. In a missing-not-at-random setting, the methods developed for the non-uniform missing mechanism could still be biased in matrix recovery. See Section 6 for a numerical example. The method we propose in this work not only deals with (a flexible subclass of) missing-not-at-random settings, but it is also applicable in the non-uniform missing settings mentioned above.

In the missing data literature, likelihood-based methods for missing-not-at-random settings commonly involve specifying a parametric distribution of the missing data mechanism. However, this assumption should be used with caution, as it is highly sensitive and may easily induce a misspecified model, resulting in biased estimation and inaccurate results. To circumvent such issues, it is preferable to adopt a missingness assumption as flexible and generally applicable as possible. This type of assumption, often referred to as an unspecified missing data mechanism [37], avoids explicitly specifying a parametric model. Instead of using the full likelihood for estimation, certain unspecified missing-not-at-random assumption allows for the derivation of a non-standard likelihood [22], which serves as the foundation for subsequent estimation. Such non-standard likelihood approaches have been used in regression analysis [32] and variable selection when confronted with informative missing [37]. One disadvantage of this approach is that not all the unknown parameters are estimable due to certain non-identification issues [16, 22].

In this work, we extend the pairwise pseudo-likelihood approach [22] to matrix completion with a mild separable informative missingness assumption (see Assumption 2.1 in Section 2), which is very flexible and generally applicable. While not all the parameters are estimable, we can identify the dispersion-scaled matrix up to a constant shift without suffering from bias due to informative missingness. This shows great promise to be applied in practice, for example, in recommendation systems where the rankings of entries are of interest.

Apart from the informative missing mechanism, our matrix completion method is based on the exponential family model, which has received extensive attention within the matrix completion literature for its efficacy in modeling non-Gaussian data, particularly discrete data. Notably, researchers have investigated its application in specific scenarios such as one-bit matrix completion [10] and multinomial matrix completion [4, 19]. The application of the exponential family model also extends to accommodating unbounded non-Gaussian observations, including Poisson matrix completion [8] and exponential family matrix completion [13, 21].

Overall, the combination of the separable missing-not-at-random mechanism and the exponential family model allows the proposed method to be applicable in a wide range of settings. We summarize the major contributions of this work as follows.

1. We formulate the pairwise pseudo-likelihood approach for matrix completion under informative missingness and exponential family model. To the best of our knowledge, the pairwise pseudo-likelihood approach has never been adopted in the matrix completion setup before. As opposed to the classical applications of pairwise pseudo-likelihood that assume i.i.d. sampling, matrix completion problems exhibit a non-identical and high-dimensional sampling structure.

2. We investigate the identifiability issues of the crucial separable missingness structures (Assumption 2.1) which lies at the core of the pairwise pseudo-likelihood approach.

3. We provide a non-trivial convergence analysis of the proposed estimator up to an identifiable equivalence class. Such analysis involves a novel concentration analysis of *matrix-valued U*-statistics where existing works on this type of concentration is sparse.

**Related Work**: To the best of our knowledge, we are one of the first works that consider the missing not at random (MNAR) setting in matrix completion and provide solid theoretical guarantees. [2] claims that they can deal with the MNAR setting. However, they assume selections and noise are independent conditioned on latent factors, as shown in their Assumption 2. On the contrary, our

setting allows missingness to depend on noise. [15] also addresses informative missingness in matrix completion. However, they require additional covariate information to complete the matrix. Compared to the above two works, our setting is more general as we do not require independence between selections and noise given the true matrix, and we do not need additional covariate information. However, we do require independence across entries given the true matrix. while [2] allows selection to be dependent among different entries. Regarding to the theoretical bounds, [2] requires additional technical conditions to develop finite sample error bounds, and their bound is point-wise, i.e., the bound is for a given location $i$. [15] also requires additional conditions on the likelihood and restricted eigenvalues to obtain convergence. Our error bound is developed under relatively weak conditions and achieves the minimax convergence rate.

## 2   Models

Let $\mathbf{A}_\star = (A_{\star,ij})_{i,j=1}^{m_1,m_2} \in \mathbb{R}^{m_1 \times m_2}$ be the matrix of interest, which is related to the observation through a generalized linear model. More specifically, we posit that the measurements $Y_{ij}$ of the $(i,j)$-th entry possesses a probability density/mass function of the exponential family form:

$$Y_{ij} \sim f_{ij}(y|\mathbf{A}_\star, \phi_\star), \qquad f_{ij}(y|\mathbf{A}_\star, \phi_\star) := h(y; \phi_\star)\exp\left(\frac{A_{\star,ij}y - G(A_{\star,ij})}{\phi_\star}\right), \qquad (1)$$

where $h : \mathbb{R} \to \mathbb{R}^+$ and $G : \mathbb{R} \to \mathbb{R}$ are the base measure function and log-partition function associated with the canonical representation, and $\phi_\star > 0$ is the dispersion parameter. Note that this family of distributions covers a wide variety of popular distributions including Bernoulli, Gaussian, and Poisson distributions. For matrix completion problems, we do not have measurements from every single entry. Let $T_{ij}$ be the observation indicator variable of the $(i,j)$-th entry, with value 1 if $Y_{ij}$ is observed and 0 otherwise. We assume that $\{((Y_{ij}, T_{ij}) : i = 1, \ldots, m_1; j = 1, \ldots, m_2\}$ are independent.

Uniform-sampling-at-random (USR) mechanism is regarded as one of the simplest missing structures for matrix completion. Under USR, $\Pr(T_{ij} = 1|Y_{ij})$ is a constant across all $i, j$, which implies that the observation indicator $T_{ij}$ is independent of the measurement $Y_{ij}$. While this has been widely used to simplify theoretical analyses in many prior ground works [e.g., 7, 6, 17], USR is a strong assumption that can be violated in many applications. To address this issue, a few analyses and methods [e.g., 31, 11, 28, 18, 4, 25, 23, 35] have been developed based on the non-uniform missing structures, where $\Pr(T_{ij} = 1|Y_{ij}) = t_{ij}$ for $0 < t_{ij} \leq 1$. Here the observation probabilities are allowed to differ across $i, j$, but the missingness remains independent of the measurement $Y_{ij}$. In this paper, we relax this restriction and allow whether an entry is observed or not to depend on the corresponding possibly unobserved measurement, leading to a challenging *missing-not-at-random* (MNAR) setup.

Matrix completion under general MNAR is ill-posed, leading to non-identifiability of $\mathbf{A}_\star$ (even under standard low-rank assumption). Indeed, general MNAR is ill-posed [33] not only in matrix completion, but also in regression [28, 31] and statistical inference [26] in general. However, some additional structure imposed within the MNAR setting can ensure identifiability. To proceed, we make the following assumption, which corresponds to a flexible subclass of MNAR settings. This assumption makes it possible to identify $\mathbf{A}_\star$ up to some equivalence relations (see Section 3).

**Assumption 2.1.** The observation probability is separable in the following sense: $\Pr(T_{ij} = 1|Y_{ij}) = t_{ij}s(Y_{ij})$, for some $t_{ij} \in (0, 1]$ and some non-negative function $s(\cdot) : \mathbb{R} \to \mathbb{R}^+$.

As will be made clear later, the proposed technique does not require the knowledge of $s(\cdot)$ and $\{t_{ij}\}$. A similar condition has been widely used in various regression problems [e.g., 22, 29, 37]. These works posit an i.i.d. setup with additional covariates, while our setup does not imply an identically distributed assumption across locations and has no covariates. Moreover, in our setup, it is not possible to observe replicates in the same location, while the i.i.d. setup generally allows replicates. Assumption 2.1 is flexible and widely applicable. Not only does it accommodate USR, it also includes non-uniform missing mechanism as a special case, where we can set $s(\cdot) \equiv 1$ and leave $\{t_{ij}\}$ variable to account for the non-uniform missing. Obviously, as the observation probability is allowed to depend on possibly unobserved $Y_{ij}$, it also includes many MNAR settings.

Clearly, we only have access to the *observed* data, i.e., $Y_{ij}$ conditional on $T_{ij} = 1$. To estimate $\mathbf{A}_\star$, we first look at the observed data likelihood of the $(i,j)$-th entry: $\Pr(Y_{ij}|T_{ij} = 1; \mathbf{A}, \phi)$ for

$\mathbf{A} \in \mathbb{R}^{m_1 \times m_2}$ and $\phi > 0$. By the Bayes' Theorem and Assumption 2.1,

$$\Pr(Y_{ij}|T_{ij} = 1; \mathbf{A}, \phi) = \frac{\Pr(T_{ij} = 1|Y_{ij})f_{ij}(Y_{ij}; \mathbf{A}, \phi)}{\int \Pr(T_{ij} = 1|Y_{ij})f_{ij}(Y_{ij}; \mathbf{A}, \phi)dY_{ij}}$$

$$= s(Y_{ij})\frac{1}{\int s(y)f_{ij}(y; \mathbf{A}, \phi)dy}f_{ij}(Y_{ij}; \mathbf{A}, \phi) = s(Y_{ij})b_{ij}(\mathbf{A}, \phi)f_{ij}(Y_{ij}|\mathbf{X}; \mathbf{A}, \phi),$$

where $b_{ij}(\mathbf{A}, \phi) = 1/\int s(y)f_{ij}(y|\mathbf{A}, \phi)dy$. We see that the conditional likelihood involves unknown functions $s(\cdot)$ and $b_{ij}(\cdot)$, which makes the estimation of $\mathbf{A}_\star$ difficult. To address this issue, we adopt a pseudo-likelihood approach [22] based on local ranks.

## 3 Pseudo-likelihood approach

Let $\mathcal{E} = (e_1, \ldots, e_n) \subseteq \{1, \ldots, m_1\} \times \{1, \ldots, m_2\}$ be a lexicographically ordered set of $n$ unique locations $\{(i, j) : T_{ij} = 1\}$. (Indeed, the specific choice of ordering does not matter.) Let the corresponding measurements be $\widetilde{\mathbf{Y}} = (\tilde{Y}_1, \ldots, \tilde{Y}_n) := (Y_{e_1}, \ldots, Y_{e_n})$ and the observation indicator be $\widetilde{\mathbf{T}} = (\tilde{T}_1, \ldots, \tilde{T}_n) := (T_{e_1}, \ldots, T_{e_n})$. We also write $\tilde{A}_k = \tilde{A}_{e_k}$ for $k = 1, \ldots, n$. We decompose the vector $\widetilde{\mathbf{Y}}$ into two vectors: the order statistics $\widetilde{\mathbf{Y}}_{(\cdot)} = (\tilde{Y}_{(1)}, \ldots, \tilde{Y}_{(n)})$ and the rank statistics $\mathbf{R} = (R_1, \ldots, R_n)$. Precisely, $\tilde{Y}_{(j)}$ is the $j$-th smallest entry in $\widetilde{\mathbf{Y}}$ and $R_k$ is the rank of the $k$-th entry in $\widetilde{\mathbf{Y}}$. To motivate the proposed pseudolikelihood in (3), we first consider the conditional likelihood based on the full rank statistics given the observed data:

$$\Pr(\mathbf{R}|\widetilde{\mathbf{Y}}_{(\cdot)}, \widetilde{\mathbf{T}} = \mathbf{1}; \mathbf{A}, \phi) = \frac{\prod_{k=1}^n s(\tilde{Y}_k)t_{e_k}f_{e_k}(\tilde{Y}_k; \mathbf{A}, \phi)}{\sum_{\pi \in \Xi} \prod_{k=1}^n s(\tilde{Y}_{(k)})t_{e_k}f_{e_k}(\tilde{Y}_{\pi(k)}; \mathbf{A}, \phi)}$$

$$= \frac{\prod_{k=1}^n \exp(\tilde{A}_k\tilde{Y}_k/\phi)}{\sum_{\pi \in \Xi} \prod_{k=1}^n \exp(\tilde{A}_k\tilde{Y}_{\pi(k)}/\phi)}, \qquad (2)$$

where $\Xi$ is the set of all one-to-one maps from $\{1, \ldots, n\}$ to $\{1, \ldots, n\}$, i.e., permutations. We notice that (2) does not involve unknown components $s(\cdot)$ and $t_{ij}$ due to the separable assumption (Assumption 2.1), and does not depend on the base measure $h(\cdot)$ and the log-partition function $G(\cdot)$. However, (2) is computationally infeasible due to the summation over all permutations. The proposed pairwise pseudo-likelihood consider local ranks for pairs of observations. For any $k$ and $k'$, let $\mathbf{R}_{kk'}^L$ denote the local rank statistic of $\tilde{Y}_k$ and $\tilde{Y}_{k'}$ among the pair $(\tilde{Y}_k, \tilde{Y}_{k'})$. We denote $\widetilde{\mathbf{Y}}_{(k,k')}^L$ as the local order statistics $(\min\{\tilde{Y}_k, \tilde{Y}_{k'}\}, \max\{\tilde{Y}_k, \tilde{Y}_{k'}\})$. Instead of the full conditional probability (2), we consider the product of all possible combinations of the local rank conditional probability on observations:

$$\prod_{k<k'} \Pr(\mathbf{R}_{kk'}^L = \mathbf{r}_{kk'}^L|\widetilde{\mathbf{Y}}_{(k,k')}^L, \tilde{T}_k = \tilde{T}_{k'} = 1; \mathbf{A}, \phi)$$

$$= \prod_{k<k'} \frac{\exp\left(\frac{\tilde{A}_k\tilde{Y}_k + \tilde{A}_{k'}\tilde{Y}_{k'}}{\phi}\right)}{\exp\left(\frac{\tilde{A}_k\tilde{Y}_k + \tilde{A}_{k'}\tilde{Y}_{k'}}{\phi}\right) + \exp\left(\frac{\tilde{A}_k\tilde{Y}_{k'} + \tilde{A}_{k'}\tilde{Y}_k}{\phi}\right)} = \prod_{k<k'} \frac{1}{1 + \exp(-(\tilde{Y}_k - \tilde{Y}_{k'})(\tilde{A}_k - \tilde{A}_{k'})/\phi)}.$$

$$\qquad (3)$$

Similar to (2), this pairwise pseudo-likelihood (3) (of $\mathbf{A}$ and $\phi$) does not contain unknown functions and quantities. However, unlike (2), it does not involve all permutations and is therefore significantly easier to compute.

The negative logarithm of the pairwise pseudo-likelihood reads

$$\sum_{1 \le k < k' \le n} \log(1 + R_{kk'}(\phi^{-1}\mathbf{A})), \qquad (4)$$

where $R_{kk'}(\phi^{-1}\mathbf{A}) = \exp\{-(\tilde{Y}_k - \tilde{Y}_{k'})(\phi^{-1}\tilde{A}_k - \phi^{-1}\tilde{A}_{k'})\}$. We notice two immediate issues with estimating $\mathbf{A}_\star$ (and $\phi_\star$) via minimizing (4).

*Scale Equivalence*: The values of (4) evaluated at any two pairs $(\mathbf{A}_1, \phi_1)$ and $(\mathbf{A}_2, \phi_2)$ are the same when $\phi_1^{-1}\mathbf{A}_1 = \phi_2^{-1}\mathbf{A}_2$. Therefore, (4) does not have the ability to distinguish between these pairs.

In other words, if $\phi_\star > 0$ is unknown, (4) would not be able to identify elements in the equivalence class of $\mathbf{A}_\star$ under equivalence relation: $\mathbf{A} \sim c_1 \mathbf{A}$ for any $c_1 > 0$. Instead, we try to estimate the dispersion-scaled matrix $\phi_\star^{-1} \mathbf{A}_\star$. Therefore, we consider

$$\ell(\mathbf{A}) = \sum_{1 \le k < k' \le n} \log(1 + R_{kk'}(\mathbf{A})).$$

However, this does not solve all the identifiability issues, and, indeed, $\ell$ cannot identify a shift-equivalence class described below.

*Shift Equivalence*: Let $\mathbf{J}$ be a matrix with all entries being one. Consider $\mathbf{A} + c_2 \mathbf{J}$ for any $c_2 \in \mathbb{R}$. Then

$$\ell(\mathbf{A} + c_2 \mathbf{J}) = R_{kk'}((\mathbf{A} + c_2 \mathbf{J})) = \exp\{-(\tilde{Y}_k - \tilde{Y}_{k'})(\tilde{A}_k + c_2 - \tilde{A}_{k'} - c_2)\}$$
$$= \exp\{-(\tilde{Y}_k - \tilde{Y}_{k'})(\tilde{A}_k - \tilde{A}_{k'})\} = \ell(\mathbf{A}).$$

Combining the scale and shift equivalence, we can only estimate $\mathbf{A}_\star$ up to an equivalence relation $\mathbf{A} \sim c_1 \mathbf{A} + c_2 \mathbf{J}$ for any $c_1 > 0$ and $c_2 \in \mathbb{R}$, which we will refer to as scale-shift equivalence. We remark that the scale-shift equivalence still allows the identification of much useful information from $\mathbf{A}_\star$, such as ranking an arbitrary set of entries of $\mathbf{A}_\star$. For example, in recommender system applications, one is mostly interested in the ranking within each row/column. Among the elements in the scale-shift equivalence class, we choose to estimate the following representer

$$\bar{\mathbf{A}}_\star = \phi_\star^{-1} \mathbf{A}_\star - \frac{\langle \mathbf{J}, \phi_\star^{-1} \mathbf{A}_\star \rangle}{\langle \mathbf{J}, \mathbf{J} \rangle} \mathbf{J} = \phi_\star^{-1} \mathbf{A}_\star - \frac{\langle \mathbf{J}, \phi_\star^{-1} \mathbf{A}_\star \rangle}{m_1 m_2} \mathbf{J}, \tag{5}$$

by imposing the constraint $\langle \mathbf{J}, \mathbf{A} \rangle = 0$ in the optimization. Here $\langle \mathbf{A}, \mathbf{B} \rangle = \sum_{i,j} A_{ij} B_{ij}$ for any matrices $\mathbf{A}, \mathbf{B} \in \mathbb{R}^{m_1 \times m_2}$.

Overall, we propose the following penalized pairwise pseudo-likelihood estimator

$$\widehat{\mathbf{A}} = \underset{\langle \mathbf{J}, \mathbf{A} \rangle = 0, \|\mathbf{A}\|_\infty \le a}{\operatorname{argmin}} \ell(\mathbf{A}) + \lambda \|\mathbf{A}\|_\star, \tag{6}$$

where $\|\mathbf{A}\|_\star$ and $\|\mathbf{A}\|_\infty (= \max_{i,j} A_{ij})$ represent the nuclear norm and the entrywise max norm of a matrix $\mathbf{A}$ respectively, and $a, \lambda \ge 0$ are tuning parameters. We also use $\|\mathbf{A}\|_F$ to denote the Frobenius norm of a matrix $\mathbf{A}$. Nuclear norm regularization has been commonly used to promote low-rankness in the estimation [25, 24, 14]. Since $\ell$ is convex, this optimization is convex. The discussion of the optimization algorithm is given in Appendix C. One natural question is whether there would be further hidden identifiability issues beyond scale-shift equivalence. In Section 5, we will provide a finite-sample error bound of the proposed estimator (6) based on the pairwise pseudo-likelihood, which indicates convergence to $\bar{\mathbf{A}}_\star$, eliminating the possibility of additional identifiability issues.

## 4 Identifiability based on separable assumption

One of the major difficulties associated with informative missing is non-identifiability. We first emphasize the non-identifiability for constant shift is not an artifact of the pseudo-likelihood approach. The root cause is the informative missingness (Assumption 2.1). Here is a simple univariate example inspired from [27] to illustrate this point. Suppose we observe from two data-generating models, whose observations are identical in distributions.

**Model I**: $Y_1 \sim \mathcal{N}(-1, 1)$ with observation probability $\Pr(T_1 = 1 | Y_1 = y) = \frac{\exp(y)}{1 + \exp(y)}$, then

$$\Pr(T_1 = 1, Y_1 = y) = p_{\mathcal{N}}(y + 1) \frac{\exp(y)}{1 + \exp(y)},$$

where $p_{\mathcal{N}}(\cdot)$ is the p.d.f. of standard normal distribution.

**Model II**: $Y_2 \sim \mathcal{N}(0, 1)$ with observation probability $\Pr(T_2 = 1 | Y_2 = y) = \exp(-1/2) \frac{\exp(-y)}{1 + \exp(-y)}$, then

$$\Pr(T_2 = 1, Y_2 = y) = p_{\mathcal{N}}(y + 1) \exp(-1) \exp(y + 1) \frac{\exp(-y)}{1 + \exp(-y)}$$

$$= p_{\mathcal{N}}(y + 1) \frac{\exp(y)}{1 + \exp(y)} = \Pr(T_1 = 1, Y_1 = y).$$

Extending to the matrix form, the observation probabilities of the following two models, where $\mathbf{Y}_1, \mathbf{Y}_2 \in \mathbb{R}^{m_1 \times m_2}$, are exactly the same. **Model I**: $\text{vec}(\mathbf{Y}_1) \sim \mathcal{N}(-\mathbf{1}, \mathbf{I})$, $t_{1,ij} = 1$ for any $(i,j)$ and $s_1(y) = \frac{\exp(y)}{1+\exp(y)}$. **Model II**: $\text{vec}(\mathbf{Y}_2) \sim \mathcal{N}(\mathbf{0}, \mathbf{I})$, $t_{2,ij} = 1$ for any $(i,j)$ and $s_2(y) = \exp(-1/2)\frac{\exp(-y)}{1+\exp(-y)}$.

As such, under Assumption 2.1, we cannot identify the constant shift. We note that, low-rank assumption generally would not provide enough additional information to eliminate this identifiability issue, as constant shift corresponds to at most a rank-1 perturbation.

The identification of the dispersion parameter is a difficult task because of the fact that at most one observation is available for each entry. Interestingly, as we have shown in the Appendix (Theorem B.2), under Assumption 2.1, the identification of the dispersion parameter is actually feasible in Gaussian distributions *with replicates*. However, it is unclear whether the dispersion parameter can be identified in a typical matrix completion setup, which often does not allow replicates. That said, previous works on exponential family matrix completion [21, 13] assume the dispersion parameter is known, under which there would not be a related identifiability issue.

## 5  Theoretical guarantee

Recall that $\boldsymbol{A}_\star \in \mathbb{R}^{m_1 \times m_2}$. We denote some convenient notation for dimensions, i.e., $m = \min\{m_1, m_2\}$, $M = \max\{m_1, m_2\}$, $d = m_1 + m_2$. We use the notation $\lesssim$ $(\gtrsim)$ to denote less (greater) than up to an absolute multiplicative constant. We write $a \asymp b$ if $a \lesssim b$ and $b \gtrsim a$. Furthermore, define $\pi_L = \min_{i \in [m_1], j \in [m_2]} \Pr(T_{ij} = 1)$ and $\pi_U = \max_{i \in [m_1], j \in [m_2]} \Pr(T_{ij} = 1)$. We use $[n]$ to represent $\{1, \ldots, n\}$ for integer $n$. In this section, we derive the convergence of $\|\widehat{\boldsymbol{A}} - \bar{\boldsymbol{A}}_\star\|_F$. Recall that $\bar{\boldsymbol{A}}_\star$, defined in (5), is the representer in the equivalence class of $\boldsymbol{A}_\star$.

**Assumption 5.1.** The following conditions hold.

(C1) There exists an absolute constant $\rho > 0$ such that $\pi_U/\pi_L \leq \rho$.

(C2) There exists a constant $B$ such that $\|\mathbf{Y}\|_\infty \leq B$ almost surely.

(C3) There exists some constant $\kappa > 0$ (where $\kappa$ can depend on $\|\bar{\boldsymbol{A}}_\star\|_\infty$) such that $\mathbb{E}(Z_{ij,i'j'}^2) \geq \kappa$ for any $i, i' \in [m_1], j, j' \in [m_2]$, where

$$Z_{ij,i'j'} = (Y_{ij} - Y_{i'j'}) \times \frac{\exp((Y_{ij} - Y_{i'j'})(A_{ij} - A_{i'j'})/2)}{1 + \exp((Y_{ij} - Y_{i'j'})(A_{ij} - A_{i'j'}))}.$$

Condition (C1) is posited to avoid some specific entries being sampled with very low probability in a relative sense, where the trace-norm penalization fails to work [11, 31]. Note that both $\pi_U$ and $\pi_L$ are allowed to diminish to zero as $m_1, m_2 \to \infty$, but Condition (C1) implies that their diminishing orders are the same. Condition (C2) is a technical assumption for analyzing the concentration inequalities of the involved $U$-statistics in pairwise pseudolikelihood. Note that this does not violate the parametric assumption on the distribution of $Y$. For example, truncated normal distribution satisfies both. We leave the extension to a light-tail type of assumption for future work. Condition (C3) is a technical condition, and is used in deriving the (expected) Hessian of the loss function with respect to $\mathbf{A}$ (see (8) in Appendix A). Note that (expected) Hessian of the loss is often important for deriving the convergence rate and so it is reasonable that a related term shows up in our condition. Indeed, this assumption Here, we provide further discussion to show that it is indeed a mild condition. Intuitively, it posits a positive *lower bound* for an expectation of a *squared* random variable. This expectation is always non-negative and is zero only when $Z_{ij,i'j'}$ is exactly zero almost everywhere. For noisy matrix completion settings, this assumption is very mild because, when there are noises, this variable is not exactly zero almost surely. Next, we show that with the exponential family model, we can explicitly characterize $\kappa$. First note that when $\|\bar{\boldsymbol{A}}_\star\|_\infty \leq a$ as assumed in Theorem 5.3 and $\|\mathbf{Y}\|_\infty \leq B$ as in Condition (C2), we have $\mathbb{E} Z_{ij,i'j'}^2 \geq \frac{\exp(4aB)}{[1+\exp(4aB)]^2} \mathbb{E}\left\{ Y_{ij}^2 + Y_{i'j'}^2 - 2Y_{ij}Y_{i'j'} \right\} \geq \frac{\exp(4aB)}{[1+\exp(4aB)]^2}[\text{Var}(Y_{ij}) + \text{Var}(Y_{i'j'})]$. Recall the density of $Y_{ij}$ (1), one can derive $\text{Var}(Y_{ij}) = G''(\boldsymbol{A}_{\star,ij})\phi$, where $G''(\cdot)$ is nonnegative, from the well-known variance formula for exponential family. Therefore one can take $\kappa = \min_{i,j} \frac{\exp(4aB)}{[1+\exp(4aB)]^2} 2\phi[G''(\boldsymbol{A}_{\star,ij})]$.

**Lemma 5.2.** *We have $\mathbb{E}\{\nabla\ell(\mathbf{A}_\star)\} = 0$, where $\mathbb{E}(\cdot)$ is the expectation under the true parameter $\mathbf{A}_\star$.*

**Theorem 5.3.** *Assume $\mathrm{rank}(\bar{A}_\star) \leq r$ and $\left\|\bar{A}_\star\right\|_\infty \leq a$ for some positive constants $r, a > 0$, under Assumptions 2.1 and 5.1, if we further assume $m\pi_U \gtrsim \log(d^2)$ and $\lambda \asymp B^2 \log(d) \left[m_1 m_2 \pi_U \sqrt{M\pi_U}\right]$, then with probability at least $1 - 6/d$, the following holds:*

$$\frac{1}{m_1 m_2} \left\|\widehat{\mathbf{A}} - \bar{A}_\star\right\|_F^2 \lesssim \max\left\{\frac{B^4 [\log d]^2}{\kappa^2} \rho^3 \frac{Mr}{m_1 m_2 \pi_L}, \frac{B^2 \log(d)}{\kappa} \rho^2 \sqrt{\frac{1}{m_1 m_2 \pi_L}}\right\}. \tag{7}$$

Our result implies that the penalized pairwise pseudolikelihood approach can consistently estimate $\bar{A}_\star$. Note that the difference between $\mathrm{rank}(\bar{A}_\star)$ and $\mathrm{rank}(A_\star)$ is at most 1. So a low-rank assumption on $A_\star$ automatically translates to a low-rank assumption on $\bar{A}_\star$. Most existing work present the upper bound concerning the number of observed entries $n$ and treat the matrix completion as a trace regression problem [e.g. 28, 18, 5, 25]. One can take $n$ as $m_1 m_2 \pi_L$ in their bound to compare their results with ours. Similar to the bound established in [18], our bound has two components and matches with the rates in their upper bound (up to some constants and logarithmic factor). [17] and [28] show a bound that has the same order as the first term (up to some constants and logarithmic factor) with some additional assumptions. [17] adopts the uniform sampling and boundedness of the condition number for $\|\bar{A}_\star\|$. [28] assumes that the sampling distribution follows a product distribution and the "spikiness ratio" (see $\alpha_{sp}$ in [28] ) is bounded. Besides the above matrix completion methods that use the nuclear norm regularization, the estimators utilizing the max-norm regularization [e.g. 5, 35] establish the same bound as the second term(up to some constants and logarithmic factor) when they assume the max-norm of $\bar{A}_\star$ is bounded. While the aforementioned methods address various missing mechanisms, it is important to emphasize that none of them can handle MNAR setting, where the missingness may depend on the observations. However, our method can tackle such informative missingness. It is interesting to see that our error bound resembles the same convergence rate as [18] (minimax optimal rate) up to a logarithmic order, despite that our setup allows MNAR mechanism.

In terms of theoretical analysis, the most notable distinction between our estimator with other existing ones lies in the objective function. The pairwise pseudo-likelihood we employ imposes unique theoretical challenges. Firstly, the gradient and Hessian are no longer as straightforward as those in the commonly used squared loss or negative log-likelihood loss (for exponential family). We carefully derive these two terms, expressing them as pairwise summations (see exact forms in Eq. (8) and Eq. (9)). Secondly, the elements in these pairwise summations are not mutually independent, posing difficulties in establishing the concentration inequality to bound them. Indeed, we need to develop corresponding theoretical tools for tackling the corresponding matrix concentration of a *matrix-valued $U$-statistics*. To address this challenge, we leverage the grouping lemma (Lemma A.5) to decouple these summations into different groups where mutual independence holds within each group. To obtain the efficient grouping, the decoupling is applied to those observed entries. Additionally, while the trace regression model provides a convenient tool for analyzing the sampling distribution, it implicitly assumes "sampling with replacement", i.e., every entry can be observed repeatedly. We adopt the framework of the Bernoulli model for the observation indicator to avoid the issue. However, theoretical analysis become more challenging. A conditional argument (see the conditional event $\mathcal{E}$ in (10)) is developed to address the discrepancy between these two frameworks. In addition, Lemma A.6 is established to marginalize the conditional event.

Finally, we remark that, while pseudo-likelihood approaches have been applied in regression analysis [32] and variable selection [37] to deal with informative missingness, such analyses mainly focus on i.i.d. design and usually make direct restricted eigenvalue condition of the (high-dimensional) Hessian matrix. In our problem, the eigenvalue condition is related to the observation probabilities. As in typical analysis of matrix completion, one is interested in the dependence on these probabilities, as they are allowed to diminish as $m_1, m_2 \to \infty$. As such, we also analyze the corresponding restricted eigenvalue bound, under the complicated grouping nature and identifiability issue. By adapting the techniques aforementioned, we provide a rigorous convergence result in non-i.i.d. design, which involves analyzing the concentration of a matrix-valued $U$-statistics (i.e., the Hessian matrix). This analysis distinguishes our work from a mere application of standard pseudo-likelihood theory, and the techniques used in the proof contribute to the field on their own merit.

# 6 Numerical experiments

We conduct the following simulation study to demonstrate the efficacy of the proposed method. We generate a $50 \times 50$ matrix $\mathbf{A}_\star$ with rank $r = 5$. The observations $Y_{ij}$ are generated from a Gaussian distribution with mean $A_{\star,ij}$ and variance $\sigma^2$ independently. In our study, we have settings with different variances $\sigma^2$. The probability of each entry being observed is related to the value of the entry itself: $\mathbb{P}(T_{ij} = 1|Y_{ij}) = 1/[1 + \exp(3Y_{ij})]$. Since the observation probability is smaller for larger $Y_{ij}$, there exists a distinctive distributional shift between the observed and unobserved entries, as shown in Figure 1

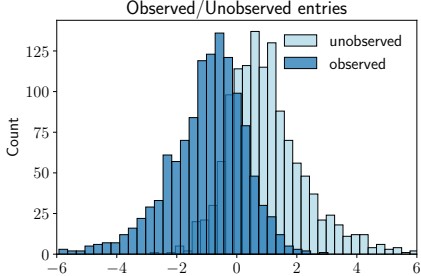

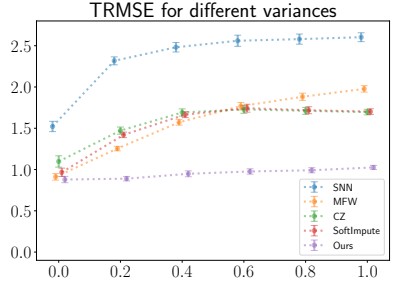

Figure 1: Observation bias with variance $\sigma^2 = 1$.

Figure 2: TRMSE with standard error for different variances $\sigma^2 = 0.0, 0.2, 0.4, 0.6, 0.8, 1.0$.

We use the observed entries as training data and equally split the unobserved data as validation and test data. We compare our method with SoftImpute [26], CZ [5], MFW [35] and SNN [2]. The validation data is used for hyper-parameter tuning in each method. Since the objective function is convex in the proposed method, we only tune the regularization parameter $\lambda$, and fix the number of iterations as $T = 100$ and step size $\eta = 1.0$ in Algorithm 1. We use the output of SoftImpute [26] with the same regularization parameter $\lambda$ as a warm-up initialization to shorten the training time. For SoftImpute [26], CZ [5] and MFW [35], we tune the hyper-parameters involved in the optimization and regularization as suggested. As for SNN [2], we choose uniform weights and spectral threshold suggested in [12], and choose the number of neighbors between 1 and 2. Due to the identifiability issue, the validation data is also used to learn a shift and scale parameter (via a simple linear regression) for the proposed method, which is then used in reporting error metrics on test data.

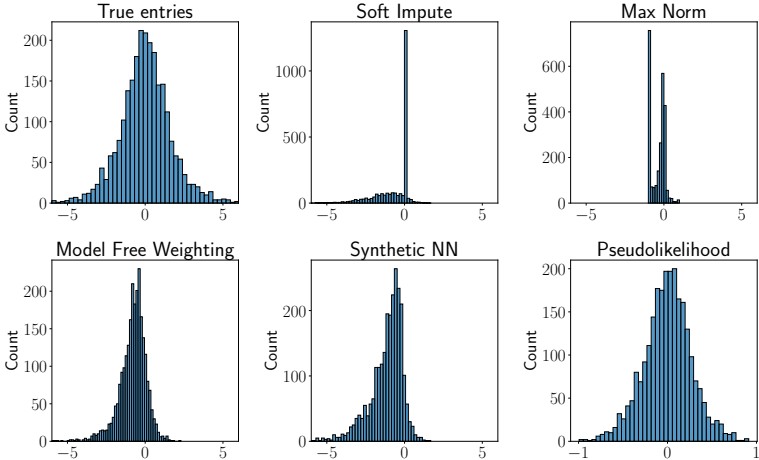

Figure 3: The recovered entries by existing methods are left skewed with $\sigma^2 = 1$.

Before getting into the error metric, a simple check on the bias of each method is via histograms. We pick one run with $\sigma^2 = 1$ and plot the distribution of recovered entries without any transformation for each method, as shown in Figure 3. It shows that only the proposed method is able to mitigate

Table 1: Computational time comparison with $\sigma^2 = 1$.

| Methods | Time (s) |
|---------|----------|
| Soft Impute | $0.01 \pm 0.005$ |
| Max Norm (CZ) | $0.22 \pm 0.09$ |
| Model Free Weighting (MFW) | $1.90 \pm 0.75$ |
| Synthetic NN (SNN) | $15.23 \pm 1.39$ |
| Pseudolikelihood | $8.67 \pm 1.23$ |

the observational bias due to the underlying informative missing structure, and exhibits a symmetric distribution, while the distributions generated by other methods are left-skewed, due to the left-skewness of distribution of the observed entries.

We added comparisons of the computational time regarding the setting in Figure 2 ($\sigma^2 = 1$). The computational times are listed in Table 1. While incorporating more complex missing mechanisms, our method and SNN also take the most time. One practical way to speed up the computation of our method is to use a stochastic version of Algorithm 1 (i.e., training in batches). The focus of this paper is more on the robust recovery when encountering informative missing, and less on the computational efficiency with the knowledge that it could be theoretically slower than other SVD-based methods. However, our method is still faster than SNN, where both methods consider more complex missing mechanisms. Given the promising statistical properties of the proposed method, a future direction is to develop scalable algorithms for the proposed estimator or its variants.

To further validate the effectiveness of the proposed method, we vary the variances $\sigma^2$ in the simulation. This setting is designed to differentiate non-uniform missingness and informative missingness. When the variance is small, the informative missingness is less severe, and non-uniform missingness might be used to approximately describe the missing mechanism. When the variance is large, the observational probability is more affected by the outcome as in a typical informative missingness setting. We choose the variances $\sigma^2 = 0.0, 0.2, 0.4, 0.6, 0.8, 1.0$. For each setting, we repeat the simulation 9 times and report the average test root mean squared errors (TRMSE) with standard errors, shown in Figure 2. We see that as the variance gets larger, there is a larger improvement in the proposed method with the design to account for informative missing over other methods. SNN [2] performs the worst when the variance is large, as it mainly borrows information on observed entries which introduces a substantial bias. It demonstrates the robustness of the proposed method in difficult settings where the missing structure is informative.

# 7   Real data application

In this section, we use three data examples to illustrate the robust performance of the proposed method. These are the Tobacco Dataset [9], Coat Shopping Dataset [30] and Yahoo! Webscope Dataset[1]. These datasets have been used in prior works for the demonstration of matrix completion methods [e.g., 2, 35]. Due to space limitation, we refer the readers to Appendix C.2 for more detailed discussions of the datasets and our analyses. Following the details of the implementation in Section 6, we report the results in Table 2. For the Coat Shopping Dataset and Yahoo! Webscope Dataset, the evaluations are based on associated test sets from the original data sources. As for the Tobacco Dataset, following [2], the missing data are randomly generated 100 times according to cigarette sales. Here is a summary of the results.

**Tabacco Dataset.** As we can see from Table 2, our method only performs worse than SNN for this MNAR dataset, with significantly smaller TRMSE than the other three methods. Note that in this synthetic missing data, the way to generate missingness is adapted from the SNN paper. When one entry is missed in Tobacco dataset, the entries in the following period are also missed. This does not satisfy the assumption of our work. So it is not surprising to see our method perform sub-optimality. However, the performance of our method still remains strong.

---

[1] https://webscope.sandbox.yahoo.com/catalog.php?datatype=r&did=3

Table 2: Test root mean squared errors (TRMSE) for Tabacoo Dataset, Coat Shopping Dataset, and Yahoo! Webscope Dataset. For Tabacoo Dataset, the average of TRMSE with standard errors (SE) in parentheses under 100 random missing data generations are presented.

| Method | Tabacco Dataset | Coat Shopping Dataset | Yahoo! Webscope Dataset (subset) |
|---|---|---|---|
| SoftImpute | 19.20 (0.28) | 1.41 | 1.84 |
| CZ | 20.45 (0.51) | 1.19 | 1.58 |
| MFW | 15.89 (0.24) | 1.07 | 1.28 |
| SNN | 12.09 (0.16) | 2.06 | 1.27 |
| Pseudolikelihood | 14.14 (0.36) | 1.20 | 1.12 |

**Coat Shopping Dataset.** As Table 2 shows, SNN performs much worse than the remaining methods for this dataset. MFW has the smallest TRMSE. Our method has smaller errors than SoftImpute and has comparable performance to CZ.

**Yahoo! Webscope Dataset.** Due to its large size and to simplify the computation, we conducted a selection procedure to reduce the size of the matrix. Please see details in Appendix C.2 about how to obtain the subset of the matrix. From Table 2, we can see that the two methods (SNN and our method) that are designed for MNAR have better performance than the remaining methods, and our method has the smallest TRMSE.

Overall, our method performs robustly well across all these three datasets. In our comparison, a few alternatives can perform very well in one example, but badly in another. For example, SNN has an excellent performance in the Tobacco Dataset while performing very poorly in the Coat Shopping Dataset. The robust performance of our method is appealing in practice, as the missing mechanism is often unknown.

# 8 Conclusion

In this paper, we tackle the matrix completion problem where missingness could depend on the possibly unobserved measurements, constituting a challenging missing-not-at-random setting. The proposed method is developed under a flexible separable missingness assumption, which allows us to develop a pairwise pseudo-likelihood approach. Corresponding identification is investigated. We also provide a non-trivial convergence analysis, as well as some numerical experiments to illustrate the efficacy of the proposed estimation. Due to the flexibility in both the missing structure (separable missingness) and measurement model (exponential family model), the proposed technique would be useful in a wide range of applications.

The grouping nature of the proposed method poses an additional burden in computation, particularly when dealing with a large number of observed entries. For future works, we consider adapting the stochastic grouping idea to reduce the computational cost and exploring its application in large-scale recommender systems.

# 9 Acknowledgements

The authors thank the reviewers for their helpful comments and suggestions. Portions of this research were conducted with the advanced computing resources provided by Texas A&M High Performance Research Computing. The work of Jiayi Wang is partly supported by the National Science Foundation (DMS-2401272). The work of Raymond K. W. Wong is partly supported by the National Science Foundation (DMS-1711952 and CCF-1934904). The work of K. C. G. Chan is partly supported by the National Science Foundation (DMS-1711952).

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

# A Proof of Theorem 5.3

We provide the proof of our theoretical results and discussion about identifiability and pseudo-likelihood approach below. The proof follows the roadmap from [18] to construct the convergence, whereas the details differ because we deal with a matrix-valued $U$-statistic type estimator. We mainly rely on Lemma S.4. from [36] to decouple the dependence in $U$-statistic structure. See Lemma A.5 in Section A.4.

We start by rewriting the pairwise pseudo-likelihood as

$$
\begin{aligned}
\ell(\mathbf{A}) &= \sum_{1 \leq k < k' \leq n} \{\psi(\tilde{Y}_{k \backslash k'} \tilde{A}_{k \backslash k'}) - \tilde{Y}_{k \backslash k'} \tilde{A}_{k \backslash k'}\} \\
&= \sum_{\substack{1 \leq i, i' \leq m_1 \\ 1 \leq j, j' \leq m_2}} T_{ij} T_{i'j'} \{\psi(Y_{ij \backslash i'j'} A_{ij \backslash i'j'}) - Y_{ij \backslash i'j'} A_{ij \backslash i'j'}\},
\end{aligned}
$$

where $\tilde{Y}_{k \backslash k'} := \tilde{Y}_k - \tilde{Y}_{k'} = Y_{ij} - Y_{i'j'} =: Y_{ij \backslash i'j'}$, and $\psi(t) = \log(1 + \exp(t))$. Simply, we obtain that $\psi'(t) = \exp(t)/\{1 + \exp(t)\}$, $\psi''(t) = \exp(t)/\{(1 + \exp(t))^2\}$ and $\psi'''(t) = \exp(t)(1 - \exp(t))/\{(1 + \exp(t))^3\}$.

Therefore, the first and second order derivatives of $\ell(\boldsymbol{A})$ are

$$
\nabla \ell(\mathbf{A}) = \sum_{1 \leq i, i' \leq m_1, 1 \leq j, j' \leq m_2} T_{i'j'} T_{ij} \times \{(\psi'(Y_{ij \backslash i'j'} A_{ij \backslash i'j'}) - 1) Y_{ij \backslash i'j'} \mathbf{E}_{ij \backslash i'j'}\}, \tag{8}
$$

and

$$
\nabla^2 \ell(\mathbf{A}) = \sum_{1 \leq i, i' \leq m_1, \leq j, j' \leq m_2} T_{i'j'} T_{ij} \times \{\psi''(Y_{ij \backslash i'j'} A_{ij \backslash i'j'}) Y_{ij \backslash i'j'}^2 \mathrm{vec}(\mathbf{E}_{ij \backslash i'j'})^{\otimes 2}\}, \tag{9}
$$

where $\mathbf{E}_{ij \backslash i'j'} = \mathbf{E}_{ij} - \mathbf{E}_{i'j'}$, $\mathbf{E}_{ij} \in \mathbb{R}^{m_1 \times m_2}$ is the canonical basis with value 1 at the $(i, j)$-th entry and 0 elsewhere, $\mathrm{vec}(\mathbf{X})$ is the standard vectorization of matrix $\mathbf{X}$ and $\mathbf{v}^{\otimes 2} = \mathbf{v}\mathbf{v}^\top$ for $\mathbf{v} \in \mathbb{R}^{m_1 m_2}$.

## A.1 Lemmas about gradient

Note that

$$
\begin{aligned}
\nabla \ell(\bar{\boldsymbol{A}}_\star) &= - \sum_{1 \leq k < k' \leq n} \frac{R_{kk'}(\bar{\boldsymbol{A}}_\star)}{1 + R_{kk'}(\bar{\boldsymbol{A}}_\star)} (\tilde{Y}_k - \tilde{Y}_{k'})(\mathbf{E}_{e_k} - \mathbf{E}_{e_{k'}}) \\
&= - \sum_{\substack{(i,j),(i',j') \in [m_1] \times [m_2] \\ (i,j) \prec (i',j')}} \frac{R_{ij,i'j'}(\bar{\boldsymbol{A}}_\star)}{1 + R_{ij,i'j'}(\bar{\boldsymbol{A}}_\star)} (\tilde{Y}_{ij} - \tilde{Y}_{i'j'})(\mathbf{E}_{ij} - \mathbf{E}_{i'j'}) T_{ij} T_{i'j'} \\
&= - \frac{1}{2} \sum_{(i,j),(i',j') \in [m_1] \times [m_2]} \frac{R_{ij,i'j'}(\bar{\boldsymbol{A}}_\star)}{1 + R_{ij,i'j'}(\bar{\boldsymbol{A}}_\star)} (\tilde{Y}_{ij} - \tilde{Y}_{i'j'})(\mathbf{E}_{ij} - \mathbf{E}_{i'j'}) T_{ij} T_{i'j'},
\end{aligned}
$$

where $(i, j) \prec (i', j')$ means $(i, j)$ appears before $(i', j')$ with the same ordering rule within $\mathcal{E}$, e.g., dictionary order.

**Proof of Lemma 5.2.** Note that

$$
\begin{aligned}
\mathbb{E}\{\nabla \ell(\bar{\boldsymbol{A}}_\star)\} = \mathbb{E}\Bigg\{ &- \sum_{\substack{(i,j),(i',j') \in [m_1] \times [m_2] \\ (i,j) \prec (i',j')}} \frac{R_{ij,i'j'}(\bar{\boldsymbol{A}}_\star)}{1 + R_{ij,i'j'}(\bar{\boldsymbol{A}}_\star)} (\tilde{Y}_{ij} - \tilde{Y}_{i'j'})(\mathbf{E}_{ij} - \mathbf{E}_{i'j'}) T_{ij} T_{i'j'} \Bigg\} \\
= &-\mathbb{E}\Bigg\{ \sum_{\substack{(i,j),(i',j') \in [m_1] \times [m_2] \\ (i,j) \prec (i',j')}} \mathbb{E}\Bigg\{ \frac{R_{ij,i'j'}(\bar{\boldsymbol{A}}_\star)}{1 + R_{ij,i'j'}(\bar{\boldsymbol{A}}_\star)} (\tilde{Y}_{ij} - \tilde{Y}_{i'j'})(\mathbf{E}_{ij} - \mathbf{E}_{i'j'}) \Big| T_{ij} = T_{i'j'} = 1 \Bigg\} \Bigg\}.
\end{aligned}
$$

The proof of $\mathbb{E}\left\{\frac{R_{ij,i'j'}(\bar{A}_\star)}{1+R_{ij,i'j'}(\bar{A}_\star)}(\tilde{Y}_{ij}-\tilde{Y}_{i'j'})(\mathbf{E}_{ij}-\mathbf{E}_{i'j'})\Big|T_{ij}=T_{i'j'}=1\right\}=0$ directly follows Theorem 4.1 from [29].

$\square$

To simplify the notation, we denote

$$\mathbf{S}_{ij,i'j'} = -\frac{R_{ij,i'j'}(\bar{A}_\star)}{1+R_{ij,i'j'}(\bar{A}_\star)}(\tilde{Y}_{ij}-\tilde{Y}_{i'j'})(\mathbf{E}_{ij}-\mathbf{E}_{i'j'})T_{ij}T_{i'j'}.$$

Recall that $m = \min\{m_1, m_2\}, d = m_1 + m_2, M = \max\{m_1, m_2\}, |Y_k| \le B$ almost surely.

**Lemma A.1.** *With the condition that $m\pi_U \gtrsim \log(d^2)$. We have*

$$\Pr\left\{\|\nabla\ell(\bar{A}_\star)\| \gtrsim \sqrt{B^2\log(d)}\left[m_1 m_2 \pi_U \sqrt{M\pi_U}\right]\right\} \le \frac{3}{d}.$$

Denote

$$C_{ij,i'j'} = -\frac{R_{ij,i'j'}(\bar{A}_\star)}{1+R_{ij,i'j'}(\bar{A}_\star)}(\tilde{Y}_{ij}-\tilde{Y}_{i'j'}).$$

First of all, we can verify that

$$\mathbb{E}(C_{ij,i'j'}T_{ij}T_{i'j'}) = \mathbb{E}\left\{\mathbb{E}[C_{ij,i'j'}T_{ij}T_{i'j'} \mid T_{ij}=1, T_{i'j'}=1]\right\} = 0.$$

Define the event

$$\mathcal{E} = \{\sum_{i,j} T_{i,j} = n \text{ with sampling matrices } \boldsymbol{X}_1,\ldots,\boldsymbol{X}_n\} \tag{10}$$

, where $\boldsymbol{X}_k = \boldsymbol{E}_{i_k,j_k}$ for some index $(i_k, j_k), k = 1,\ldots,n$. Without loss of generality, we consider the case when $n$ is even. Therefore, by Lemma A.5, we have the following decomposition.

$$\nabla\ell(\mathbf{A}) \mid \mathcal{E} = \sum_{g=1}^{n-1}\sum_{(k,k')\in G_g} C_{k,k'}[\mathbf{X}_k - \mathbf{X}_{k'}],$$

where $C_{k,k'} = C_{i_k,j_k,i_{k'},j_{k'}}$. Within every group $G_g$, there is no repeated index. Therefore, every element in the group $G_g$ is independent of other elements in $G_g$ conditioned on $\mathcal{E}$.

$$\mathbb{E}(C_{k,k'} \mid \mathcal{E}) = 0, \qquad \forall k,k' = 1,\ldots,n.$$

Next, we use Matrix Bernstein's inequality to bound $\|\sum_{(k,k')\in G_g} C_{k,k'}[\mathbf{X}_k - \mathbf{X}_{k'}]\|$ conditioned on the event $\mathcal{E}$.

Take $\boldsymbol{S}_{k,k'} = C_{k,k'}[\mathbf{X}_k - \mathbf{X}_{k'}]$ and we have

$$\left\|\mathbb{E}\sum_{(k,k')\in G_g}\boldsymbol{S}_{k,k'}\boldsymbol{S}_{k,k'}^\mathsf{T} \mid \mathcal{E}\right\|$$

$$= \left\|\mathbb{E}[\sum_{(k,k')\in G_g} C_{k,k'}^2[\mathbf{X}_k - \mathbf{X}_{k'}][\mathbf{X}_k - \mathbf{X}_{k'}]^\mathsf{T} \mid \mathcal{E}]\right\|$$

$$\le B^2 \left\|\sum_{(k,k')\in G_g}[\mathbf{X}_k - \mathbf{X}_{k'}][\mathbf{X}_k - \mathbf{X}_{k'}]^\mathsf{T}\right\|$$

$$\le B^2 \left(\left\|\sum_{(k,k')\in G_g}\boldsymbol{X}_k\boldsymbol{X}_k^\mathsf{T} + \boldsymbol{X}_{k'}\boldsymbol{X}_{k'}^\mathsf{T}\right\| + \left\|\sum_{(k,k')\in G_g}\boldsymbol{X}_k\boldsymbol{X}_{k'}^\mathsf{T} + \boldsymbol{X}_{k'}\boldsymbol{X}_k^\mathsf{T}\right\|\right).$$

Take $i_k$ and $j_k$ as the corresponding row index and column index for $\boldsymbol{X}_k$ such that $\boldsymbol{X}_k = \boldsymbol{E}_{i_k,j_k}$.

$$\left\|\sum_{(k,k')\in G_g}\boldsymbol{X}_k\boldsymbol{X}_k^\mathsf{T} + \boldsymbol{X}_{k'}\boldsymbol{X}_{k'}^\mathsf{T}\right\| = \left\|\sum_{k=1}^n \boldsymbol{E}_{i_k,i_k}\right\| = 1.$$

The last inequality is due to the fact that the diagonal matrices have the operator norm 1.

$$\left\| \sum_{(k,k')\in G_g} \boldsymbol{X}_k \boldsymbol{X}_{k'}^\mathsf{T} + \boldsymbol{X}_{k'} \boldsymbol{X}_k^\mathsf{T} \right\| \le \sum_{(k,k')\in G_g} \| \boldsymbol{X}_k \boldsymbol{X}_{k'}^\mathsf{T} + \boldsymbol{X}_{k'} \boldsymbol{X}_k^\mathsf{T} \| \le \sum_{(k,k')\in G_g} 2\mathbb{1}\{j_k = j_{k'}\}.$$

Then

$$\left\| \mathbb{E} \sum_{(k,k')\in G_g} \boldsymbol{S}_{k,k'} \boldsymbol{S}_{k,k'}^\mathsf{T} \mid \mathcal{E} \right\| \le B^2 \left[ 1 + \sum_{(k,k')\in G_g} 2\mathbb{1}\{j_k = j_{k'}\} \right]$$

Similarly,

$$\left\| \mathbb{E} \sum_{(k,k')\in G_g} \boldsymbol{S}_{k,k'}^\mathsf{T} \boldsymbol{S}_{k,k'} \mid \mathcal{E} \right\| \le B^2 \left[ 1 + \sum_{(k,k')\in G_g} 2\mathbb{1}\{i_k = i_{k'}\} \right]$$

Take

$$\xi_{n,g} = \max\{ \sum_{(k,k')\in G_g} 2\mathbb{1}\{j_k = j_{k'}\}, \sum_{(k,k')\in G_g} 2\mathbb{1}\{i_k = i_{k'}\}\}. \tag{11}$$

Then by Theorem 6.1 from [34], we have

$$\Pr\left\{ \left\| \sum_{(k,k')\in g} \mathbf{S}_{k,k'} \right\| \ge c\sqrt{B^2(1 + \xi_{n,g})[2\log(d) + 2\log(m_1 m_2)]} \mid \mathcal{E} \right\}$$

$$\le d\exp\{-[2\log(m_1 m_2) + 2\log(d)]\} = d\left( \frac{1}{(m_1 m_2)^2 d^2} \right) = \frac{1}{(m_1 m_2)^2 d},$$

for some constant $c > 0$.

Note that $\left\| \nabla \ell(\bar{\boldsymbol{A}}_\star) \right\| \mid \mathcal{E} \le \sum_{g=1}^{n-1} \left\| \sum_{(k,k')\in g} \mathbf{S}_{k,k'} \right\|$. By applying the union bound over all the groups, we obtain

$$\Pr\left\{ \left\| \nabla \ell(\bar{\boldsymbol{A}}_\star) \right\| \ge c \sum_{g=1}^{n-1} \sqrt{B^2(1 + \xi_{n,g})[2\log(d) + 2\log(m_1 m_2)]} \mid \mathcal{E} \right\} \le n\frac{1}{(m_1 m_2)^2 d} \le \frac{1}{(m_1 m_2)d}.$$

Marginalize the event $\mathcal{E}$ and we have

$$\Pr\left\{ \left\| \nabla \ell(\bar{\boldsymbol{A}}_\star) \right\| \ge c \sum_{g=1}^{n-1} \sqrt{B^2(1 + \xi_{n,g})[2\log(d) + 2\log(m_1 m_2)]} \right\} \le \frac{1}{(m_1 m_2)d}.$$

Note that

$$\sum_{g=1}^{n-1} \sqrt{B^2(1 + \xi_{n,g})[2\log(d) + 2\log(m_1 m_2)]}$$

$$\le \sum_{g=1}^{n-1} \sqrt{B^2[2\log(d) + 2\log(m_1 m_2)]} + \sum_{g=1}^{n-1} \sqrt{B^2 \xi_{n,g}[2\log(d) + 2\log(m_1 m_2)]}$$

$$\le (n-1)\sqrt{B^2[2\log(d) + 2\log(m_1 m_2)]} + \sqrt{n-1}\sqrt{B^2[2\log(d) + 2\log(m_1 m_2)](\sum_{g=1}^{G} \xi_{n,g})}$$

$$\le n\sqrt{B^2[2\log(d) + 2\log(m_1 m_2)]} + \sqrt{n}\sqrt{B^2[2\log(d) + 2\log(m_1 m_2)](\sum_{g=1}^{G} \xi_{n,g})}.$$

Therefore, we have

$$\Pr\left\{\left\|\nabla\ell(\bar{\boldsymbol{A}}_\star)\right\| \geq c\left[n\sqrt{B^2[2\log(d)+2\log(m_1m_2)]} + \sqrt{n}\sqrt{B^2[2\log(d)+2\log(m_1m_2)](\sum_{g=1}^{G}\xi_{n,g})}\right]\right\}$$
$$\leq \frac{1}{(m_1m_2)d}.$$

By Lemma A.6, we adopt the bound for $n$ and $\sum_{g=1}^{n-1}\xi_{n,g}$ and further obtain that

$$\Pr\left\{\left\|\nabla\ell(\bar{\boldsymbol{A}}_\star)\right\| \geq c\sqrt{B^2[2\log(d)+2\log(m_1m_2)]}\left[3m_1m_2\pi_U + \sqrt{3m_1m_2\pi_U}\sqrt{2m_1m_2M\pi_U^2}\right]\right\}$$
$$\leq \frac{1}{(m_1m_2)d} + \exp(-m_1m_2\pi_L) + 2M\exp(-m_1m_2\pi_U^2/2).$$
$$\Pr\left\{\left\|\nabla\ell(\bar{\boldsymbol{A}}_\star)\right\| \gtrsim \sqrt{B^2[\log(d)+\log(m_1m_2)]}\left[m_1m_2\pi_U(1+\sqrt{M\pi_U})\right]\right\}$$
$$\leq \frac{1}{(m_1m_2)d} + \exp(-m_1m_2\pi_L) + 2M\exp(-m_1m_2\pi_U^2/2).$$

With the condition that $m\pi_U \gtrsim \log(d^2)$. We have

$$\Pr\left\{\left\|\nabla\ell(\bar{\boldsymbol{A}}_\star)\right\| \gtrsim \sqrt{B^2[\log(d)+\log(m_1m_2)]}\left[m_1m_2\pi_U\sqrt{M\pi_U}\right]\right\} \leq \frac{3}{d}.$$

The conclusion follows by assimilating universal constants in $\gtrsim$.

## A.2 Lemmas about Hessian

Recall that $\psi(t) = \log(1+\exp(t))$, $\psi'(t) = \exp(t)/\{1+\exp(t)\}$, $\psi''(t) = \exp(t)/\{(1+\exp(t))^2\}$ and $\psi'''(t) = \exp(t)(1-\exp(t))/\{(1+\exp(t))^3\}$. It is simple to verify $|\psi'(t)| \leq 1$, $|\psi''(t)| \leq 0.25$ and $|\psi'''(t)| \leq 0.1$.

The Hessian matrix reads

$$\nabla^2\ell(\mathbf{A}) = \sum_{1\leq k<k'\leq n}\{\psi''(Y_{ij\backslash i'j'}A_{ij\backslash i'j'})Y_{ij\backslash i'j'}^2\text{vec}(\mathbf{E}_{ij\backslash i'j'})^{\otimes 2}\}$$

$$= \sum_{1\leq k<k'\leq n}(Y_k - Y_{k'})^2\frac{\exp((\tilde{Y}_k - \tilde{Y}_{k'})(\tilde{A}_k - \tilde{A}_{k'}))}{(1+\exp((\tilde{Y}_k - \tilde{Y}_{k'})(\tilde{A}_k - \tilde{A}_{k'})))^2}\text{vec}(\mathbf{E}_k - \mathbf{E}_{k'})^{\otimes 2}$$

$$= \frac{1}{2}\sum_{(i,j),(i',j')\in[m_1]\times[m_2]}T_{ij}T_{i'j'}(Y_{ij} - Y_{i'j'})^2\frac{\exp((Y_{ij} - Y_{i'j'})(A_{ij} - A_{i'j'}))}{(1+\exp((Y_{ij} - Y_{i'j'})(A_{ij} - A_{i'j'})))^2}\text{vec}(\mathbf{E}_{ij} - \mathbf{E}_{i'j'})^{\otimes 2}$$

$$= \frac{1}{2}\sum_{(i,j),(i',j')\in[m_1]\times[m_2]}T_{ij}T_{i'j'}Z_{ij,i'j'}^2\text{vec}(\mathbf{E}_{ij} - \mathbf{E}_{i'j'})^{\otimes 2},$$

where

$$Z_{ij,i'j'} = (Y_{ij} - Y_{i'j'})\frac{\exp((Y_{ij} - Y_{i'j'})(A_{ij} - A_{i'j'})/2)}{1+\exp((Y_{ij} - Y_{i'j'})(A_{ij} - A_{i'j'}))}.$$

**Lemma A.2.** *For any* $\mathbf{U} \in \mathbb{R}^{m_1\times m_2}$ *such that* $\langle\mathbf{J},\mathbf{U}\rangle = 0$, *and assume that* $\mathbb{E}(Z_{ij,i'j'}^2) \geq \kappa, \forall(i,j),(i',j')\in[m_1]\times[m_2]$ *when* $\|\mathbf{A}\|_\infty \leq a$ *for some constant* $a > 0$, *we have that*

$$\mathbb{E}\text{vec}(\mathbf{U})^\top\nabla^2\ell(\mathbf{A})\text{vec}(\mathbf{U}) \geq \kappa m_1m_2\pi_L^2\|\mathbf{U}\|_F^2.$$

*Proof.* We derive the quadratic form first and take the expectation to obtain that

$$
\begin{aligned}
\mathbb{E}\mathbf{u}^\top \nabla^2 \ell(\mathbf{A})\mathbf{u} &= \frac{1}{2} \sum_{(i,j),(i',j')\in[m_1]\times[m_2]} \mathbb{E}\{T_{ij}T_{i'j'}Z_{ij,i'j'}^2(U_{ij}-U_{i'j'})^2\} \\
&\geq \frac{1}{2}\kappa \sum_{(i,j),(i',j')\in[m_1]\times[m_2]} \mathbb{E}\{T_{ij}T_{i'j'}(U_{ij}-U_{i'j'})^2\} \\
&\geq \frac{1}{2}\kappa\pi_L^2 \sum_{(i,j),(i',j')\in[m_1]\times[m_2]} (U_{ij}-U_{i'j'})^2 \\
&= \frac{1}{2}\kappa\pi_L^2 \sum_{(i,j),(i',j')\in[m_1]\times[m_2]} U_{ij}^2 + U_{i'j'}^2 - 2U_{ij}U_{i'j'} \\
&= \kappa m_1 m_2 \pi_L^2 \|\mathbf{U}\|_F^2 .
\end{aligned}
$$

$\square$

We now start to lower bound $\mathbf{u}^\top \nabla^2 \ell(\mathbf{A})\mathbf{u}$. We consider the following constraint set

$$
\mathcal{C} = \left\{ \mathbf{U} \in \mathbb{R}^{m_1\times m_2} \ \middle|\ \langle \mathbf{J}, \mathbf{U}\rangle = 0 \right\},
$$

Still, we derive the argument by conditioning on the event $\mathcal{E}$ defined in (10).

$$
\operatorname{vec}(\mathbf{U})^\top \nabla^2 \ell(\mathbf{A})\operatorname{vec}(\mathbf{U}) \mid \mathcal{E} = \frac{1}{2} \sum_{k,k'=1,\dots,n} Z_{i_k j_k, i_{k'} j_{k'}}^2 (U_{i_k j_k} - U_{i_{k'} j_{k'}})^2.
$$

where $(i_k, j_k)$ is the entry where $\boldsymbol{E}_{i_k,j_k}=1$.

By Lemma A.5, WLOG, we assume $n$ is even. We decompose the summation into

$$
\operatorname{vec}(\mathbf{U})^\top \nabla^2 \ell(\mathbf{A})\operatorname{vec}(\mathbf{U}) \mid \mathcal{E} = \sum_{g=1}^{n-1} \sum_{(k,k')\in G_g} Z_{i_k j_k, i_{k'} j_{k'}}^2 (U_{i_k j_k} - U_{i_{k'} j_{k'}})^2,
$$

where within every group $G_g$, there is no repeated index. Denote

$$
\Sigma_g = \sum_{(k,k')\in G_g} \varepsilon_{k,k'} Z_{i_k j_k, i_{k'} j_{k'}} (\boldsymbol{X}_k - \boldsymbol{X}_{k'}).
$$

where $\varepsilon_{k,k'}$ are independent Radamacher variables. For convenience, we denote $\mathbf{u} = \operatorname{vec}(\mathbf{U}) \in \mathbb{R}^{m_1 m_2}$ for any $\mathbf{U} \in \mathbb{R}^{m_1\times m_2}$.

**Lemma A.3.** *For all $\mathbf{U} \in \mathcal{C}$, the following holds*

$$
\left|\mathbf{u}^\top \nabla^2 \ell(\mathbf{A})\mathbf{u} - \mathbb{E}\mathbf{u}^\top \nabla^2 \ell(\mathbf{A})\mathbf{u}\right| \gtrsim B^2\|\boldsymbol{U}\|_* \sqrt{\log(d)} \left[m_1 m_2 \pi_U \sqrt{M\pi_U}\right] + B^2 \log(d)(m_1 m_2 \pi_U)^2 (m_1 m_2 \pi_L)^{-1/2}
$$

*with probability at least $1 - 3/d$.*

*Proof.* Denote
$$
\mathcal{C}_T := \{\boldsymbol{U} \in \mathcal{C} : \|\boldsymbol{U}\|_* \leq T\}.
$$
$$
W_T = \sup_{\mathbf{U}\in\mathcal{C}(T)} \left|\mathbf{u}^\top \nabla^2 \ell(\mathbf{A})\mathbf{u} - \mathbb{E}\mathbf{u}^\top \nabla^2 \ell(\mathbf{A})\mathbf{u}\right|,
$$

Let $v = \sqrt{m\pi_U/\pi_L}$. By Lemma A.4, we have with probability at least $1 - 1/d$,

$$
\left|\mathbf{u}^\top \nabla^2 \ell(\mathbf{A})\mathbf{u} - \mathbb{E}\mathbf{u}^\top \nabla^2 \ell(\mathbf{A})\mathbf{u}\right| \gtrsim B^2 \log(d)(m_1 m_2 \pi_U)^2 (m_1 m_2 \pi_L)^{-1/2}.
$$

Next, we focus on the case when $\|U\|_* \geq v$. We will show that the probability of the following bad event is small

$$
\mathcal{B} = \left\{ \exists \mathbf{U} \in \mathcal{C} \text{ such that } \left|\mathbf{u}^\top \nabla^2 \ell(\mathbf{A})\mathbf{u} - \mathbb{E}\mathbf{u}^\top \nabla^2 \ell(\mathbf{A})\mathbf{u}\right| \gtrsim B^2\|\boldsymbol{U}\|_* \sqrt{\log(d)} \left[m_1 m_2 \pi_U \sqrt{M\pi_U}\right], \|\boldsymbol{U}\|_* \geq v \right\}.
$$

We use a standard peeling argument. For $l \in \mathbb{N}$ set

$$\mathcal{S}_l = \left\{ \mathbf{U} \in \mathcal{C}(r) : \alpha^l \nu \le \|\mathbf{U}\|_* \le \alpha^{l+1}\nu \right\}.$$

For each $T > \nu$, define the following event

$$\mathcal{B}_l = \left\{ \exists \mathbf{U} \in \mathcal{S}_l \text{ such that } \left| \mathbf{u}^\top \nabla^2 \ell(\mathbf{A})\mathbf{u} - \mathbb{E}\mathbf{u}^\top \nabla^2 \ell(\mathbf{A})\mathbf{u} \right| \gtrsim B^2 \|\mathbf{U}\|_* \sqrt{\log(d)} \left[ m_1 m_2 \pi_U \sqrt{M\pi_U} \right] \right\}.$$

By lemma A.4, with probability smaller than $(m_1 m_2) \exp\left\{ \frac{-T^2 \pi_L}{m\pi_U} \log(d^3) \right\}$, we have that

$$W_T \ge B^2 T \sqrt{\log(d)} \left[ m_1 m_2 \pi_U \sqrt{M\pi_U} \right].$$

Therefore, we obtain that

$$\Pr(\mathcal{B}_l) \le (m_1 m_2) \exp\left\{ -\frac{(\alpha^{2l} v^2)\pi_L}{m\pi_U} \log(d^3) \right\}.$$

Using the union bound, we have

$$\begin{aligned}
\Pr(\mathcal{B}) &\le \sum_{l=0}^{\infty} \Pr(\mathcal{B}_l) \le \sum_{l=1}^{\infty} (m_1 m_2) \exp\left\{ \frac{-(\alpha^{2l} v^2)\pi_L}{m\pi_U} \log(d^3) \right\} + 1/d \\
&\le \sum_{l=1}^{\infty} (m_1 m_2) \exp\left\{ \frac{-(v^2)\pi_L}{m\pi_U} \log(d^3) 2l \log(\alpha) \right\} + 1/d \\
&\le \int_{l=1}^{\infty} (m_1 m_2) \exp\left\{ \frac{-(v^2)\pi_L}{m\pi_U} \log(d^3) 2l \log(\alpha) \right\} + 1/d \\
&\le (m_1 m_2) \exp\{-2\log(\alpha)\log(d^3)\} + 1/d \le 2/d.
\end{aligned}$$

Combine two cases, and the conclusion follows. $\qquad\square$

We now make up the concentration property on $W_T$.

**Lemma A.4.** *Conditioned on event $\mathcal{E}$, For any $T > \nu$, we denote*

$$\mathcal{C}(T) := \{ \mathbf{U} \in \mathcal{C}(r) : \|\mathbf{U}\|_* \le T \} \quad and \quad W_T := \sup_{\mathbf{U} \in \mathcal{C}(T)} \left| \mathbf{u}^\top \nabla^2 \ell(\mathbf{A})\mathbf{u} - \mathbb{E}\mathbf{u}^\top \nabla^2 \ell(\mathbf{A})\mathbf{u} \right|.$$

*When $T \le \sqrt{m\pi_U/\pi_L}$, we have*

$$\mathbb{P}\left( W_T \gtrsim B^2 \log(d)(m_1 m_2 \pi_U)^2 (m_1 m_2 \pi_L)^{-1/2} \right) \le 1/d.$$

*When $T \ge \sqrt{m\pi_U/\pi_L}$, By taking $t = \frac{T\sqrt{3\log d}\sqrt{M\pi_U}}{m_1 m_2 \pi_U}$, we have*

$$\mathbb{P}\left( W_T \gtrsim B^2 T \sqrt{\log(d)} \left[ m_1 m_2 \pi_U \sqrt{M\pi_U} \right] \right) \le (m_1 m_2) \exp\left\{ \frac{-T^2 \pi_L}{m\pi_U} \log(d^3) \right\}.$$

*Proof.* Still, without generality, we focus on the case when $n$ is even. Recall that conditioned on event $\mathcal{E}$, we have

$$W_T = \sup_{\mathbf{U} \in \mathcal{C}(r,T)} \left| \mathbf{u}^\top \nabla^2 \ell(\mathbf{A})\mathbf{u} - \mathbb{E}\mathbf{u}^\top \nabla^2 \ell(\mathbf{A})\mathbf{u} \right|$$

$$= \sup_{\mathbf{U} \in \mathcal{C}(r,T)} \left| \sum_g \sum_{(k,k') \in G_g} Z^2_{i_k j_k, i_{k'} j_{k'}} (U_{i_k j_k} - U_{i_{k'} j_{k'}})^2 - \mathbb{E}\sum_g \sum_{(k,k') \in G_g} Z^2_{i_k j_k, i_{k'} j_{k'}} (U_{i_k j_k} - U_{i_{k'} j_{k'}})^2 \right|$$

$$\le \sum_g \sup_{\mathbf{U} \in \mathcal{C}(r,T)} \left| \sum_{(k,k') \in G_g} Z^2_{i_k j_k, i_{k'} j_{k'}} (U_{i_k j_k} - U_{i_{k'} j_{k'}})^2 - \mathbb{E} \sum_{(k,k') \in G_g} Z^2_{k,k'} (U_{i_k j_k} - U_{i_{k'} j_{k'}})^2 \right|,$$

and

$$Z_{g,T} = \sup_{\mathbf{U} \in \mathcal{C}(r,T)} \left| \sum_{(k,k') \in G_g} Z^2_{i_k j_k, i_{k'} j_{k'}} (U_{i_k j_k} - U_{i_{k'} j_{k'}})^2 - \mathbb{E} \sum_{(k,k') \in G_g} Z^2_{i_k j_k, i_{k'} j_{k'}} (U_{i_k j_k} - U_{i_{k'} j_{k'}})^2 \right|.$$

Note that conditioned on the event $\mathcal{E}$, within each grouping $G_g$, every term $Z^2_{i_k j_k, i_{k'} j_{k'}} (U_{i_k j_k} - U_{i_{k'} j_{k'}})^2$ is independent of the others. The standard symmetrization trick still applies here,

$$\mathbb{E} Z_{g,T} \mid \mathcal{E} \leq 2\mathbb{E} \sup_{\mathbf{U} \in \mathcal{C}(T)} \left| \sum_{(k,k') \in G_g} \varepsilon_{k,k'} Z^2_{i_k j_k, i_{k'} j_{k'}} (U_{i_k j_k} - U_{i_{k'} j_{k'}})^2 \right|.$$

Since $|Z_{i_k i_{k'}}| \leq 2B$ and $\|\mathbf{U}\|_\infty = 1$, $Z^2_{i_k j_k, i_{k'} j_{k'}} (U_{i_k j_k} - U_{i_{k'} j_{k'}})^2 \leq 16B^2$ for every $(k,k')$. Therefore, $\phi(u) = u^2$, $|\phi(u) - \phi(v)| \leq |u+v||u-v| \leq 8B|u-v|$. The contraction inequality yields

$$\mathbb{E} Z_{g,T} \mid \mathcal{E} \leq 2\mathbb{E} \sup_{\mathbf{U} \in \mathcal{C}(T)} \left| \sum_{(k,k') \in G_g} \varepsilon_{k,k'} Z^2_{i_k j_k, i_{k'} j_{k'}} (U_{i_k j_k} - U_{i_{k'} j_{k'}})^2 \right|$$

$$\leq 16B\mathbb{E} \sup_{\mathbf{U} \in \mathcal{C}(T)} \left| \sum_{(k,k') \in G_g} \varepsilon_{k,k'} Z_{i_k j_k, i_{k'} j_{k'}} (U_{i_k j_k} - U_{i_{k'} j_{k'}}) \right|$$

$$\leq 16B\mathbb{E} \left( \sup_{\mathbf{U} \in \mathcal{C}(T)} \left| \sum_{(k,k') \in G_g} \varepsilon_{k,k'} Z_{i_k j_k, i_{k'} j_{k'}} \langle \mathbf{X}_k - \mathbf{X}_{k'}, \mathbf{U} \rangle \right| \mid \mathcal{E} \right)$$

$$\leq 16B\mathbb{E} \left( \sup_{\mathbf{U} \in \mathcal{C}(T)} \left| \left\langle \sum_{(k,k') \in G_g} \varepsilon_{k,k'} Z_{i_k j_k, i_{k'} j_{k'}} (\mathbf{X}_k - \mathbf{X}_{k'}), \mathbf{U} \right\rangle \right| \mid \mathcal{E} \right)$$

$$\leq 16B\mathbb{E} \left( \left\| \sum_{(k,k') \in G_g} \varepsilon_{k,k'} Z_{i_k j_k, i_{k'} j_{k'}} (\mathbf{X}_k - \mathbf{X}_{k'}) \right\| \mid \mathcal{E} \right) \sup_{\mathbf{U} \in \mathcal{C}(T)} \|\mathbf{U}\|_\star$$

$$\leq 16BT\mathbb{E} \left( \|\Sigma_g\| \mid \mathcal{E} \right).$$

Take $\gamma_{k,k'}(U) = Z^2_{i_k j_k, i_{k'} j_{k'}} (U_{i_k j_k} - U_{i_{k'} j_{k'}})^2 - \mathbb{E} Z^2_{i_k j_k, i_{k'} j_{k'}} (U_{i_k j_k} - U_{i_{k'} j_{k'}})^2$ Note that

$$|Z^2_{i_k j_k, i_{k'} j_{k'}} (U_{i_k j_k} - U_{i_{k'} j_{k'}})^2| \leq 16B^2.$$

Then

$$\mathbb{E} \gamma_{k,k}(U) = 0$$
$$\sup_{(k,k')} \sup_{U \in \mathcal{C}(T)} \gamma_{k,k'}(U)/(32B^2) \leq 1$$

By Massart's concentration inequality (e.g., Theorem 14.2 in [3]). We have that conditioning on the event $\mathcal{E}$, for any $t > 0$,

$$\Pr \left( \left| \sup_{U \in \mathcal{C}(T)} \frac{1}{n/2} \sum_{(k,k') \in G_g} \gamma_{k,k'}(U)/(32B^2) \right| > \mathbb{E} \left| \sup_{U \in \mathcal{C}(T)} \frac{1}{n/2} \sum_{(k,k') \in G_g} \gamma_{k,k'}(U)/(32B^2) \right| + t \right)$$
$$\leq \exp(-(n/2)t^2/8)$$

$$\Pr \left( \left| \sup_{U \in \mathcal{C}(T)} \frac{1}{n/2} \sum_{(k,k') \in G_g} \gamma_{k,k'}(U) \right| > \mathbb{E} \left| \sup_{U \in \mathcal{C}(T)} \frac{1}{n/2} \sum_{(k,k') \in G_g} \gamma_{k,k'}(U) \right| + 32B^2 t \right) \leq \exp(-(n/2)t^2/8)$$

$$\Pr \left( Z_{g,T} > \mathbb{E} Z_{g,T} + 16nB^2 t \right) \leq \exp(-nt^2/16).$$

Apply a union bound, we have

$$\Pr\left(W_T > \sum_{g=1}^{n-1} \mathbb{E}Z_{g,T} + 16n(n-1)B^2t \mid \mathcal{E}\right) \le (n-1)\exp(-nt^2/16).$$

Next, we marginalize the event $\mathcal{E}$.

$$\Pr\left(W_T > \sum_{g=1}^{n-1} \mathbb{E}Z_{g,T} + 16n^2B^2t\right) \le \mathbb{E}[n\exp(-nt^2/16)] \le (m_1 m_2)\mathbb{E}[\exp(-nt^2/16)]$$

$$\le (m_1 m_2)\exp(-\mathbb{E}(n)t^2/16) \le (m_1 m_2)\exp(-m_1 m_2 \pi_L t^2/16).$$

Next, we considering bounding $\sum_{g=1}^{n-1} \mathbb{E}Z_{g,T}$. Note that

$$\sum_{g=1}^{n-1} \mathbb{E}Z_{g,T} \le 16BT\mathbb{E}\left[\mathbb{E}\left(\sum_{g=1}^{n-1} \|\Sigma_g\| \mid \mathcal{E}\right)\right].$$

By using a similar argument in Lemma A.1, we are able to show that for any $x > 0$,

$$\Pr\left\{\sum_{g=1}^{n-1} \|\Sigma_g\| \gtrsim x\sqrt{B^2\log(d)}\left[m_1 m_2 \pi_U \sqrt{M\pi_U}\right]\right\} \le \frac{3}{d}\exp\{-x\log(d^2)\}.$$

Therefore,

$$\sum_{g=1}^{n-1} \mathbb{E}Z_{g,T} \lesssim 16B^2T\sqrt{\log(d)}\left[m_1 m_2 \pi_U \sqrt{M\pi_U}\right].$$

Combining the result from Lemma A.6, we have

$$\Pr\left(W_T \gtrsim B^2T\sqrt{\log(d)}\left[m_1 m_2 \pi_U \sqrt{M\pi_U}\right] + (m_1 m_2 \pi_U)^2 B^2 t\right) \le (m_1 m_2)\exp(-m_1 m_2 \pi_L t^2/16).$$

When $T \le \sqrt{m\pi_U/\pi_L}$, we have

$$\Pr\left(W_T \gtrsim B^2\log(d)(m_1 m_2 \pi_U)^2(m_1 m_2 \pi_L)^{-1/2}\right) \le 1/d.$$

When $T \ge \sqrt{m\pi_U/\pi_L}$, By taking $t = \frac{T\sqrt{3\log d}\sqrt{M\pi_U}}{m_1 m_2 \pi_U}$, we have

$$\Pr\left(W_T \gtrsim B^2T\sqrt{\log(d)}\left[m_1 m_2 \pi_U \sqrt{M\pi_U}\right]\right) \le (m_1 m_2)\exp\left\{\frac{-T^2\pi_L}{m\pi_U}\log(d^3)\right\}.$$

$\square$

### A.3 Proof of Theorem 5.3

*Proof.* It follows from the definition of the estimator $\widehat{\mathbf{A}}$ that

$$\ell(\widehat{\mathbf{A}}) + \lambda\left\|\widehat{\mathbf{A}}\right\|_\star \le \ell(\bar{\mathbf{A}}_\star) + \lambda\left\|\bar{\mathbf{A}}_\star\right\|_\star,$$

equivalently

$$\ell(\widehat{\mathbf{A}}) - \ell(\bar{\mathbf{A}}_\star) \le \lambda\left(\left\|\bar{\mathbf{A}}_\star\right\|_\star - \left\|\widehat{\mathbf{A}}\right\|_\star\right)$$

which implies

$$\langle\nabla\ell(\bar{\mathbf{A}}_\star), \text{vec}(\widehat{\mathbf{A}} - \bar{\mathbf{A}}_\star)\rangle + \text{vec}(\widehat{\mathbf{A}} - \bar{\mathbf{A}}_\star)^\top \nabla^2\ell(\widetilde{\mathbf{A}})\text{vec}(\widehat{\mathbf{A}} - \bar{\mathbf{A}}_\star) \le \lambda\left(\left\|\bar{\mathbf{A}}_\star\right\|_\star - \left\|\widehat{\mathbf{A}}\right\|_\star\right).$$

where $\widetilde{\mathbf{A}} = t\bar{\mathbf{A}}_\star + (1-t)\widehat{\mathbf{A}}$ for some $t \in [0,1]$.

Let's denote $\boldsymbol{\Delta} = \widehat{\mathbf{A}} - \bar{\mathbf{A}}_\star$.

From Lemma A.3 and Lemma A.2, we have

$$\mathbf{u}^\top \nabla^2 \ell(\mathbf{A})\mathbf{u} \gtrsim \mathbb{E}\left\{\mathbf{u}^\top \nabla^2 \ell(\mathbf{A})\mathbf{u}\right\} - B^2 \|\boldsymbol{U}\|_* \sqrt{\log(d)}\left[m_1 m_2 \pi_U \sqrt{M\pi_U}\right] - B^2 \log(d)(m_1 m_2 \pi_U)^2 (m_1 m_2 \pi_L)^{-1/2}$$

$$\gtrsim \kappa m_1 m_2 \pi_L^2 \|\boldsymbol{U}\|_F^2 - B^2 \|\boldsymbol{U}\|_* \sqrt{\log(d)}\left[m_1 m_2 \pi_U \sqrt{M\pi_U}\right] - B^2 \log(d)(m_1 m_2 \pi_U)^2 (m_1 m_2 \pi_L)^{-1/2}.$$

with probability at most $1 - 3/d$.

Therefore with probability at most $1 - 3/d$,

$$\kappa m_1 m_2 \pi_L^2 \|\boldsymbol{\Delta}\|_F^2 \lesssim B^2 \|\boldsymbol{\Delta}\|_* \sqrt{\log(d)}\left[m_1 m_2 \pi_U \sqrt{M\pi_U}\right] + B^2 \log(d)(m_1 m_2 \pi_U)^2 (m_1 m_2 \pi_L)^{-1/2}$$

$$+ \mathrm{vec}(\widehat{\mathbf{A}} - \bar{\boldsymbol{A}}_\star)^\top \nabla^2 \ell(\widetilde{\mathbf{A}})\mathrm{vec}(\widehat{\mathbf{A}} - \bar{\boldsymbol{A}}_\star)$$

$$\lesssim B^2 \|\boldsymbol{\Delta}\|_* \sqrt{\log(d)}\left[m_1 m_2 \pi_U \sqrt{M\pi_U}\right] + B^2 \log(d)(m_1 m_2 \pi_U)^2 (m_1 m_2 \pi_L)^{-1/2}$$

$$+ \lambda\left(\left\|\bar{\boldsymbol{A}}_\star\right\|_\star - \left\|\widehat{\mathbf{A}}\right\|_\star\right) + \left\|\nabla \ell(\bar{\boldsymbol{A}}_\star)\right\| \|\boldsymbol{\Delta}\|_\star.$$

Let $\{\mathbf{u}_k \in \mathbb{R}^{m_1}\}$ and $\{\mathbf{v}_k \in \mathbb{R}^{m_2}\}$ be the left and right singular vectors of $\bar{\boldsymbol{A}}_\star$ respectively. For any matrix $\mathbf{A} \in \mathbb{R}^{m_1 \times m_2}$, we let $\mathrm{row}(\mathbf{A}) \subseteq \mathbb{R}^{m_2}$ and $\mathrm{col}(\mathbf{A}) \subseteq \mathbb{R}^{m_1}$ denote its row space and column space respectively. Let the column and row span of $\bar{\boldsymbol{A}}_\star$ be $\mathcal{U}_\star = \mathrm{col}(\bar{\boldsymbol{A}}_\star) = \mathrm{span}\{\mathbf{u}_k\}$ and $\mathcal{V}_\star = \mathrm{row}(\bar{\boldsymbol{A}}_\star) = \mathrm{span}\{\mathbf{v}_k\}$ respectively. Define

$$\mathcal{M} := \{\mathbf{A} : \mathrm{row}(\mathbf{A}) \subseteq \mathcal{V}_\star, \mathrm{col}(\mathbf{A}) \subseteq \mathcal{U}_\star\},$$

$$\overline{\mathcal{M}}^\perp := \{\mathbf{A} : \mathrm{row}(\mathbf{A}) \subseteq \mathcal{V}_\star^\perp, \mathrm{col}(\mathbf{A}) \subseteq \mathcal{U}_\star^\perp\}.$$

It is easy to see that $\mathcal{M} \subseteq \overline{\mathcal{M}}$, but $\mathcal{M} \neq \overline{\mathcal{M}}$. The subspace compatibility of $\overline{\mathcal{M}}$ is upper bounded by $\sqrt{2r}$, i.e.,

$$\sup_{\mathbf{A} \in \overline{\mathcal{M}}\backslash\{0\}} \frac{\|\mathbf{A}\|_\star}{\|\mathbf{A}\|_F} \leq \sqrt{2r}.$$

We observe that

$$\left\|\widehat{\mathbf{A}}\right\|_\star = \left\|\bar{\boldsymbol{A}}_\star + \boldsymbol{\Delta}_{\overline{\mathcal{M}}} + \boldsymbol{\Delta}_{\overline{\mathcal{M}}^\perp}\right\|_\star \geq \left\|\bar{\boldsymbol{A}}_\star + \boldsymbol{\Delta}_{\overline{\mathcal{M}}^\perp}\right\|_\star - \left\|\boldsymbol{\Delta}_{\overline{\mathcal{M}}}\right\|_\star = \left\|\bar{\boldsymbol{A}}_\star\right\|_\star + \left\|\boldsymbol{\Delta}_{\overline{\mathcal{M}}^\perp}\right\|_\star - \left\|\boldsymbol{\Delta}_{\overline{\mathcal{M}}}\right\|_\star.$$

By choosing $\lambda \gtrsim 2(\left\|\nabla \ell(\bar{\boldsymbol{A}}_\star)\right\| + B^2 \sqrt{\log(d)}\left[m_1 m_2 \pi_U \sqrt{M\pi_U}\right])$, we have

$$\kappa m_1 m_2 \pi_L^2 \|\boldsymbol{\Delta}\|_F^2$$

$$\lesssim \left(\left\|\nabla \ell(\bar{\boldsymbol{A}}_\star)\right\| + B^2 \sqrt{\log(d)}\left[m_1 m_2 \pi_U \sqrt{M\pi_U}\right]\right)\left(\left\|\boldsymbol{\Delta}_{\overline{\mathcal{M}}^\perp}\right\|_\star + \left\|\boldsymbol{\Delta}_{\overline{\mathcal{M}}}\right\|_\star\right)$$

$$+ \lambda(\left\|\boldsymbol{\Delta}_{\overline{\mathcal{M}}}\right\|_\star - \left\|\boldsymbol{\Delta}_{\overline{\mathcal{M}}^\perp}\right\|_\star) + B^2 \log(d)(m_1 m_2 \pi_U)^2 (m_1 m_2 \pi_L)^{-1/2}$$

$$\lesssim 3\lambda \left\|\boldsymbol{\Delta}_{\overline{\mathcal{M}}}\right\|_\star + B^2 \log(d)(m_1 m_2 \pi_U)^2 (m_1 m_2 \pi_L)^{-1/2} \leq 3\lambda\sqrt{2r}\|\boldsymbol{\Delta}\|_F + B^2 \log(d)(m_1 m_2 \pi_U)^2 (m_1 m_2 \pi_L)^{-1/2}$$

Then, we could derive

$$\|\boldsymbol{\Delta}\|_F^2 \lesssim \frac{3\lambda\sqrt{2r}}{\kappa m_1 m_2 \pi_L^2}\|\boldsymbol{\Delta}\|_F + \frac{B^2 \log(d)(m_1 m_2 \pi_U)^2 (m_1 m_2 \pi_L)^{-1/2}}{\kappa m_1 m_2 \pi_L^2}$$

$$\frac{1}{m_1 m_2}\|\boldsymbol{\Delta}\|_F \lesssim \max\left\{\frac{\lambda^2 r}{m_1 m_2 (\kappa m_1 m_2 \pi_L^2)^2}, \frac{B^2 \log(d)}{\kappa}\left(\frac{\pi_U}{\pi_L}\right)^2 \sqrt{\frac{1}{m_1 m_2 \pi_L}}\right\}$$

Due to Lemma A.1,

$$\Pr\left\{\|\nabla \ell(\mathbf{A})\| \gtrsim \sqrt{B^2[\log(d) + \log(m_1 m_2)]}\left[m_1 m_2 \pi_U \sqrt{M\pi_U}\right]\right\} \leq \frac{3}{d}.$$

we can take $\lambda \asymp (B^2 \log(d)\left[m_1 m_2 \pi_U \sqrt{M\pi_U}\right])$, and with probability at least $1 - 6/d$, we have

$$\frac{1}{m_1 m_2}\|\boldsymbol{\Delta}\|_F^2 \lesssim \max\left\{\frac{B^4[\log d]^2}{\kappa^2}\left(\frac{\pi_U}{\pi_L}\right)^3 \frac{Mr}{m_1 m_2 \pi_L}, \frac{B^2 \log(d)}{\kappa}\left(\frac{\pi_U}{\pi_L}\right)^2 \sqrt{\frac{1}{m_1 m_2 \pi_L}}\right\}$$

$\square$

### A.4 Auxiliary lemmas

**Lemma A.5.** *For any collection of individual index pairs* $\{(j, j') : 1 \le j < j' \le n\}$,

    *(a) (From Lemma S.4. in [36]) When $n$ is even, we can decompose it into $(n-1)$ groups such that within each group, there are $n/2$ pairs and no repeated individuals.*

    *(b) When $n$ is odd, we can decompose it into $n$ groups such that within each group, there are $(n-1)/2$ pairs and no repeated individuals.*

*Proof.* The proof for part (a) is done in [36].

For part (b), when $n$ is odd, we consider an extra index $n+1$ and add all the pairs $\{(j, n+1) : 1 \le j \le n\}$ to the original collection. For the new collection of individual index pairs $\{(j, j') : 1 \le j < j' \le n+1\}$, since $n+1$ is even, we can apply part (a) and get $n$ groups such that within each group, there are $(n+1)/2$ pairs and no repeated individuals. Therefore, every index appears in each group exactly once. In each group, we remove the pair with $n+1$ in it. We now obtain $n$ groups such that within each group, there are $(n-1)/2$ pairs and no repeated individuals for the collection of individual index pairs $\{(j, j') : 1 \le j < j' \le n\}$. $\qquad\square$

Recall that

$$Z_{ij,i'j'} = (Y_{ij} - Y_{i'j'})^2 \frac{\exp((Y_{ij} - Y_{i'j'})(A_{ij} - A_{i'j'})/2)}{1 + \exp((Y_{ij} - Y_{i'j'})(A_{ij} - A_{i'j'}))},$$

$$\Sigma_g = \sum_{(ij,i'j') \in g} \varepsilon_{ij,i'j'} T_{ij} T_{i'j'} Z_{ij,i'j'} (\mathbf{E}_{ij} - \mathbf{E}_{i'j'}),$$

for any collection of non-overlap index pairs.

We analyze the upper bound for $\mathbb{E} \|\Sigma_g\|$ to prove Corollary 5.3.

**Lemma A.6.** *Take $n = \sum_{i,j} T_{i,j}$ and $\xi_{n,g}$ defined in (11). We have*

$$\Pr\left(n \ge e(\sum_{i,j} \pi_{i,j})\right) \le \exp(-m_1 m_2 \pi_L).$$

$$\Pr\left(\sum_{g=1}^{n-1} \xi_{n,g} > 2m_1 m_2 M \pi_U^2\right) \le 2M \exp\left\{-m_1 m_2 \pi_U^2 / 2\right\}.$$

*Proof.* Note that $T_{i,j}$ are independent Bernoulli random variables and $\mathbb{E}n = \sum_{i,j} \pi_{i,j}$. We apply Chernoff's inequality and obtain

$$\Pr(n \ge t) \le \exp\{-\sum_{i,j} \pi_{i,j}\} \left(\frac{e(\sum_{i,j} \pi_{i,j})}{t}\right)^t,$$

for any $t > \mathbb{E}n$. Take $t = e(\sum_{i,j} \pi_{i,j})$ and we have

$$\Pr\left(n \ge e(\sum_{i,j} \pi_{i,j})\right) \le \exp\{-\sum_{i,j} \pi_{i,j}\} \le \exp(-m_1 m_2 \pi_L).$$

Note that

$$\sum_{g=1}^{n-1} \xi_{n,g} = \max\left\{\sum_{k \ne k'}^{n} \mathbb{1}\{i_k = i_{k'}\}, \sum_{k \ne k'}^{n} \mathbb{1}\{j_k = j_{k'}\}\right\}$$

$$= \max\left\{\sum_{i=1}^{m_1} \sum_{j \ne j'}^{m_2} T_{ij} T_{ij'}, \sum_{j=1}^{m_2} \sum_{i \ne i'}^{m_1} T_{ij} T_{i'j'}\right\}.$$

We consider bounding $\sum_{i=1}^{n} \sum_{j \neq j'}^{m_2} T_{i,j} T_{i,j'}$. Similarly, without loss of generality, we assume $j$ is even, and by Lemma A.5, we can decompose

$$\sum_{i=1}^{m_1} \sum_{j \neq j'}^{m_2} T_{i,j} T_{i,j'} = \sum_{g=1}^{m_2-1} (\sum_{i=1}^{m_1} \sum_{(j,j') \in G'_g} T_{ij} T_{ij'}).$$

Within every group $G'_g$, every pair $T_{i,j} T_{i,j'}$ is independent of others. Then we apply Bernstein inequality to bound $\sum_{i=1}^{m_1} \sum_{(j,j') \in G'_g} T_{i,j} T_{i,j'}$.

$$\Pr \left( \sum_{i=1}^{m_1} \sum_{(j,j') \in G'_g} T_{ij} T_{ij'} - \sum_{i=1}^{m_1} \sum_{(j,j') \in G'_g} \pi_{i,j} \pi_{i,j'} \geq t \right) \leq \exp \left\{ \frac{-t^2/2}{\sum_{i=1}^{m_1} \sum_{(j,j') \in G'_g} \mathbb{E} T_{ij}^2 T_{ij'}^2 + t} \right\}$$

$$\leq \exp \left\{ \frac{-t^2/2}{\pi_U^2 m_1 m_2/2 + t} \right\}.$$

Take $t = m_1 m_2 \pi_U^2$, we have

$$\Pr \left( \sum_{i=1}^{m_1} \sum_{(j,j') \in G'_g} T_{ij} T_{ij'} \geq 2 m_1 m_2 \pi_U^2 \right) \leq \exp \left\{ -m_1 m_2 \pi_U^2/3 \right\}. \tag{12}$$

Take a union bound over $g = 1, \ldots, m_2 - 1$, we have

$$\Pr \left( \sum_{i=1}^{m_1} \sum_{j \neq j'}^{m_2} T_{ij} T_{ij'} > 2 m_1 m_2 (m_2 - 1) \pi_U^2 \right) \leq m_2 \exp \left\{ -m_1 m_2 \pi_U^2/3 \right\}.$$

$$\Pr \left( \sum_{i=1}^{m_1} \sum_{j \neq j'}^{m_2} T_{ij} T_{ij'} > 2 m_1 m_2 M \pi_U^2 \right) \leq M \exp \left\{ -m_1 m_2 \pi_U^2/3 \right\}.$$

With the same argument, we have

$$\Pr \left( \sum_{j=1}^{m_2} \sum_{i \neq i'}^{m_1} T_{ij} T_{ij'} > 2 m_1 m_2 M \pi_U^2 \right) \leq M \exp \left\{ -m_1 m_2 \pi_U^2/3 \right\}.$$

And therefore

$$\Pr \left( \sum_{g=1}^{n-1} \xi_{n,g} > 2 m_1 m_2 M \pi_U^2 \right) \leq 2M \exp \left\{ -m_1 m_2 \pi_U^2/3 \right\}.$$

$\square$

## B   Identifiability of dispersion parameter in Gaussian distributions

Assume $Y_{ij} \sim \mathcal{N}(\mu_{ij}, \sigma^2)$ with missing mechanism:

$$\Pr(T_{ij} = 1 | Y_{ij} = y) = c_{ij} s(y), \tag{13}$$

where $c_{ij}, s(\cdot) \in [0, 1]$.

**Lemma B.1.** *Assume that*

$$\lim_{y \to \infty} \frac{c_{ij} s(y)}{c'_{ij} s'(y)} = b_{ij} \text{ or } \lim_{y \to -\infty} \frac{c_{ij} s(y)}{c'_{ij} s'(y)} = b'_{ij},$$

*where $b_{ij}, b'_{ij}$ can not be both 0 or $\infty$ simultaneously.*

*If $\sigma \neq \sigma'$, then for any $(c_{ij}, c'_{ij}, s(y), s'(y))$, at least one of the following two statements holds:*

$$\lim_{y \to +\infty} \frac{\phi\left(\frac{y-\mu_{ij}}{\sigma}\right)}{\phi\left(\frac{y-\mu'_{ij}}{\sigma'}\right)} \frac{c_{ij}s(y)}{c'_{ij}s'(y)} = +\infty \text{ or } 0 \tag{14}$$

$$\lim_{y \to -\infty} \frac{\phi\left(\frac{y-\mu_{ij}}{\sigma}\right)}{\phi\left(\frac{y-\mu'_{ij}}{\sigma'}\right)} \frac{c_{ij}s(y)}{c'_{ij}s'(y)} = -\infty \text{ or } 0. \tag{15}$$

*Proof.* We observe that

$$\frac{\phi\left(\frac{y-\mu_{ij}}{\sigma}\right)}{\phi\left(\frac{y-\mu'_{ij}}{\sigma'}\right)} \frac{c_{ij}s(y)}{c'_{ij}s'(y)} = \exp\left\{ \frac{(\sigma^2 - \sigma'^2)y^2}{2\sigma^2\sigma'^2} + \frac{(\sigma'^2\mu - \sigma^2\mu')y}{\sigma^2\sigma'^2} + \frac{\sigma^2\mu'^2 - \sigma'^2\mu^2}{2\sigma^2\sigma'^2} \right\} \frac{c_{ij}s(y)}{c'_{ij}s'(y)},$$

With our assumption, assume $\lim_{y \to \infty} \frac{c_{ij}s(y)}{c'_{ij}s'(y)} = b_{ij}$ and $\lim_{y \to -\infty} \frac{c_{ij}s(y)}{c'_{ij}s'(y)} = b'_{ij}$.

When $b_{ij} \in (0, \infty)$, if $\sigma > \sigma'$, when $y \to \infty$, (14) converges to $\infty$.. If $\sigma < \sigma'$, when $y \to \infty$, (14) converges to $0$.

When $b_{ij} = 0$ and $b'_{ij} \neq 0$, if $\sigma < \sigma'$, when $y \to \infty$, (14) converges to $0$ is still true. If $\sigma > \sigma'$, when $y \to -\infty$, (15) converges to $\infty$. as $b'_{ij} \neq 0$.

When $b_{ij} = \infty$ and $b'_{ij} \neq \infty$, if $\sigma > \sigma'$, when $y \to \infty$, (14) converges to $\infty$. still holds. If $\sigma < \sigma'$, when $y \to -\infty$, (15) converges to $0$, as $b'_{ij} \neq \infty$. $\qquad \square$

**Theorem B.2.** *Assume at most one of $\lim_{y \to -\infty} s(y) = 0$ and $\lim_{y \to \infty} s(y) = 0$ is true, $\sigma^2$ is identifiable.*

*Proof.* Proof by contradiction. Suppose that there are two sets of parameters satisfying the same observed distribution:

$$\frac{1}{\sigma} \phi\left(\frac{y - \mu_{ij}}{\sigma}\right) c_{ij}s(y) = \frac{1}{\sigma'} \phi\left(\frac{y - \mu'_{ij}}{\sigma'}\right) c'_{ij}s'(y).$$

Therefore,

$$\frac{\phi\left(\frac{y-\mu_{ij}}{\sigma}\right)}{\phi\left(\frac{y-\mu'_{ij}}{\sigma'}\right)} \frac{c_{ij}s(y)}{c'_{ij}s'(y)} = \frac{\sigma}{\sigma'} \in (0, \infty).$$

However, if $\sigma \neq \sigma'$, by Lemma B.1, we know that the left-side will converge to $0$ or $\infty$, which violates the above equation. Thus, we must have $\sigma = \sigma'$. $\sigma^2$ is identifiable.

$\qquad \square$

# C  Algorithm and experiments

Note that the objective function and the constraint set are both convex, and the constraint set is a closed convex set. So (6) is a convex optimization problem. To deal with the constraint on $A$, one can use the Alternating Direction Method of Multipliers (ADMM) to tackle it. However, the computation of ADMM can be slow in practice. We propose a practically more efficient algorithm based on the idea of proximal gradient descent with an additional projection as detailed in Algorithm 1. The code is publicly available on GitHub[2]. In the algorithm, POCS is the projection onto the intersection of two convex sets $\{A : \langle J, A \rangle = 0\}$ and $\{A : \|A\|_\infty \leq a\}$ (see Algorithm 2). The notation $\mathcal{S}_\lambda(\cdot)$ is the soft-thresholding operator defined by $\mathcal{S}_\lambda(A) = U D_\lambda V^\top$, where $D_\lambda = \text{diag}[(d_1 - \lambda)_+, \dots, (d_r - \lambda)_+]$ with $t_+ = \max(t, 0)$, and $U \text{diag}[d_1, \dots, d_r] V^\top$ is the singular value decomposition of $A$.

---

**Algorithm 1** Projected gradient descent

---

**Initialize:** Initialize $\mathbf{A}(0)$ randomly, set learning rate $\eta$.
**for** $t = 0$ to $T$ **do**
$\quad \mathbf{K}(t) = \mathbf{A}(t) - \eta \nabla \ell(\mathbf{A}(t))$
$\quad \mathbf{Q}(t) = \mathcal{S}_\lambda(\mathbf{K}(t))$
$\quad \mathbf{A}(t+1) = \text{POCS}(\mathbf{Q}(t))$
**end for**

---

---

**Algorithm 2** POCS

---

**Initialize:** Input matrix $Q \in \mathbb{R}^{m_1 \times m_2}$. $t = 0$.
**while** $Q' \neq Q$ **do** $Q = Q'$
$\quad \tilde{Q} = Q - \left( \frac{1}{m_1 m_2} \sum_{i=1}^{m_1} \sum_{j=1}^{m_2} Q_{i,j} \right) \boldsymbol{J}$
$\quad Q'_{i,j} = \mathbb{1}\left( |\tilde{Q}_{i,j}| \leq \alpha \right) \tilde{Q}_{i,j} + \mathbb{1}\left( |\tilde{Q}_{i,j}| > \alpha \right) \text{sign}\left( \tilde{Q}_{i,j} \right) \alpha$
**end while**

---

Due to space constraints, we present the test mean absolute errors (TMAE) curve (Figure 4) for simulation setting in Section 6. Note that the trend is consistent on both TRMSE and TMAE with multiple runs.

### C.1 More simulation: missing on small entries

We conduct more simulation studies in higher dimensions with a different missing mechanism where the observation probability reads

$$\Pr(T_{ij} = 1 | Y_{ij}) = \frac{1}{1 + \exp(-3(Y_{ij} - 2))},$$

which means larger potential observations imply higher observation probabilities, as shown in Figure 5. We generate a $100 \times 100$ matrix $\mathbf{A}_\star$ with rank $r = 5$. The observations $Y_{ij}$ are generated from a Gaussian distribution with mean $A_{\star,ij}$ and variance 1 independently. Note that compared with experiments shown in Section 6, the dimension is larger and the observation probability is flipped.

As shown in Figure 6, other methods suffer from a severe observation bias, and the recovered entries are right-skewed, whereas the distribution of true entries is symmetric. Our method alleviates the observation bias and recovers the symmetric pattern of the distribution on recovered entries. The test root mean squared errors (TRMSE) and test mean absolute errors (TMAE) of recovered entries are reported in Table 3. The experiments are repeated with 9 runs, and the standard deviation of both metrics is included. Our method shows a significant advantage when the observation bias persists.

---

[2]https://github.com/jiangyuan-li/mc-w-pseudolikelihood

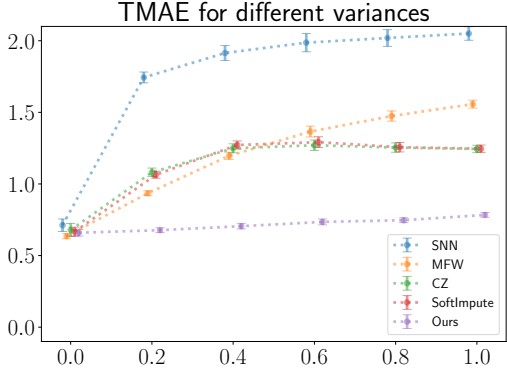

Figure 4: TMAE with standard error for different variances $\sigma^2 = 0.0, 0.2, 0.4, 0.6, 0.8, 1.0$.

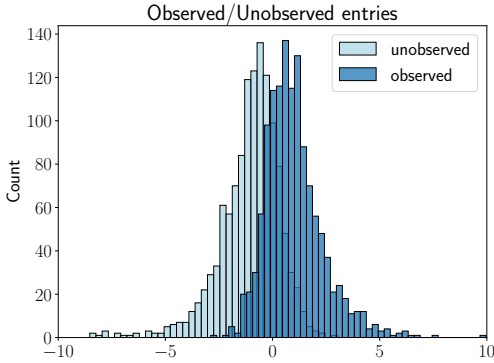

Figure 5: Distribution of observed and unobserved entries, implying the existence of observation bias.

| Method | TRMSE | TMAE |
|---|---|---|
| SoftImpute | $1.3973 \pm 0.0844$ | $1.014 \pm 0.0636$ |
| CZ | $1.410 \pm 0.0938$ | $1.0177 \pm 0.0752$ |
| MFW | $2.6763 \pm 0.0864$ | $2.2405 \pm 0.0728$ |
| SNN | $5.8402 \pm 5.5596$ | $2.7677 \pm 0.1323$ |
| Our method | $0.6482 \pm 0.0396$ | $0.4920 \pm 0.0237$ |

Table 3: Test root mean squared errors (TRMSE) and test mean absolute errors (TMAE) with standard deviations.

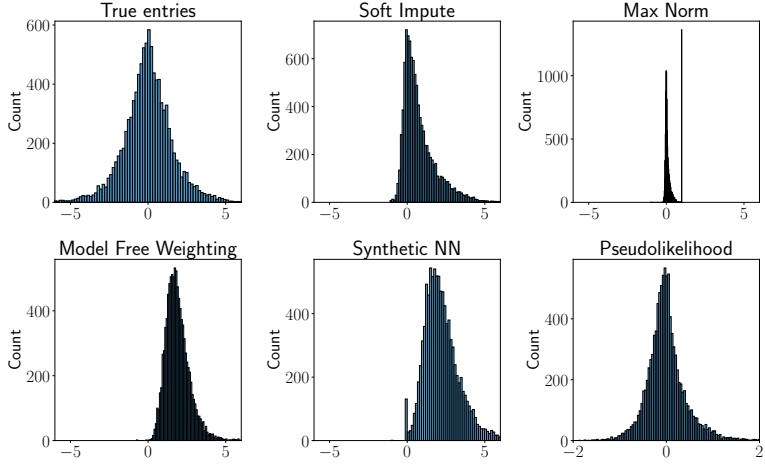

Figure 6: The recovered entries are right skewed from other methods.

## C.2 Real data application: more details

In this section, we use three data examples to illustrate the robust performance of the proposed method. Similar to the implementation procedures in Section 6, we equally separate half of the testing data points and use them as the validation dataset to tune the hyperparameters for all the methods mentioned in Section 6 and learn the shift and scale parameters for the proposed method. Then we used the remaining half of the test data points to evaluate their performance.

### C.2.1 Tobacco Dataset

This dataset is available in Table 11 in [9] and has been widely studied for synthetic control methods [e.g. 1]. From Table 11 in [9], we can obtain the Tax-Paid Per Capita Sales in Number of Packs for 51 states across the US from the year 1950 to the year 2014. California implemented a large-scale tobacco control program in 1988 and we consider it as the "treated" state. We take the remaining 50 states as the control states. For the detailed background, we refer readers to [1] and [2].

We collect data from the year 1970 to the year 2000 and restrict our focus to the 50 control states, which results in a 50 by 31 matrix. Following the same experimental setup in [2], we generate MNAR data. We introduce "interventions" to a subset of states in 1989 based on their change in the mean of cigarette sales during 1989-2000 versus that during 1970-1988. See details of the intervention probabilities in Section 6.3 in [2]. As long as an intervention is adopted for state $i$, all sales under control after 1988 are unobserved, i.e. $T_{i,j} = 0$ for $j > 19$. And we take $Y_{i,j}, j > 19$ as the test data points.

Table 2 provides the results for five methods under 100 randomizations on the intervention based on the intervention probability for every state. As we can see, our method only performs worse than SNN for this MNAR dataset, with significantly smaller TRMSE than the other three methods. Note that in order to compare the errors, we need to perform a transformation on our estimated matrix. For this study, the untransformed data can also provide valuable information about the trend of sales change for every state during these 30 years. For example, as illustrated in Figure 7, using untransformed estimated results from our method, we are able to capture the overall trend of the sales change for state KS across 30 years. Our method can capture the increasing trend of sales for the state KS after the year 1988.

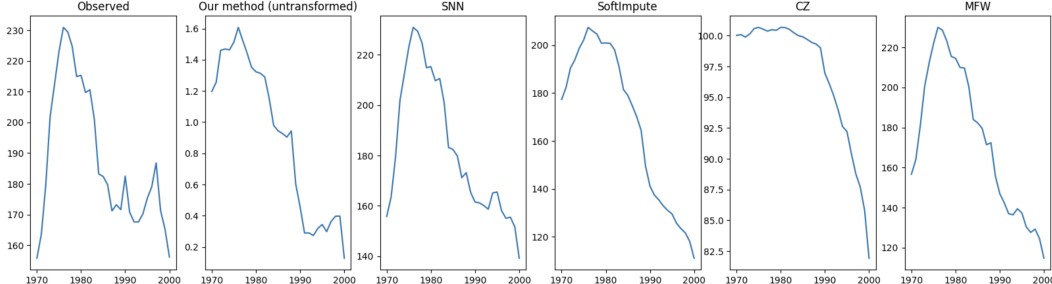

Figure 7: The first plot shows the observed sales for State KS across 30 years. The second plot is the untransformed estimated sales by our method from 1970 to 2000. The rest four plots are the estimated sales from SNN, SoftImpute, CZ, and MFW from 1970 to 2000 for the state KS.

### C.2.2 Coat Shopping Dataset

This dataset is available at `https://www.cs.cornell.edu/~schnabts/mnar/`. It contains ratings from 290 Turkers on an inventory of 300 items [30]. The training set contains 6960 self-selected ratings and the test set consists of 4640 entries. This dataset has been used as an illustration for the nonuniform missingness [e.g. 30, 35].

As Table 2 shows, SNN performs a lot worse than the remaining methods for this dataset. MFW has the smallest TRMSE. Our method has smaller errors than SoftImpute and has comparable performance to CZ.

### C.2.3 Yahoo! Webscope Dataset

This dataset is available at `https://webscope.sandbox.yahoo.com/catalog.php?datatype=r&did=3`, which contains ratings from 15,400 users on 1000 songs. The IDs for users and songs are randomly assigned. The training set includes approximately 300,000 ratings from these 15,400 users. The ratings from the training set are collected during the normal use of Yahoo! Music services for each user. The test set was constructed by surveying the first 5,400 users. And each surveyed user provides ratings for exactly 10 additional songs.

Due to its large size and to simplify the computation, we conducted a selection procedure to reduce the size of the matrix. First, we focus on the users with ID 1-50, 5401-5550, and songs with ID 1-250, which results in a matrix where we have $50$ surveyed users and $150$ unsurveyed users. Next, we construct the training (test) matrix from the original training (test) dataset with selected user IDs and selected song IDs. Then, we remove those users and songs that have no single observation in the training matrix. In the end, we obtain a matrix with $199$ users and $219$ songs.

From Table 2, we can see that the two methods (SNN and our method) that are designed for MNAR have better performance than the remaining methods, and our method has the smallest TRMSE.

