# OpenReview forum: "A Pairwise Pseudo-likelihood Approach for Matrix Completion with Informative Missingness"
_NeurIPS.cc/2024/Conference — NeurIPS 2024 spotlight_

### Official Review · Reviewer_CJYv · 2024-06-28

**Soundness:** 3
**Presentation:** 3
**Contribution:** 4
**Rating:** 7
**Confidence:** 3

**Summary:**

This paper describes a method for estimating missing values of a given non-completed matrix in a situation called MNAR, where the probability distribution for the observation depends on both the matrix index and the matrix value.

They propose a nonparametric method to prevent bias due to model selection. While missing value estimation in MNAR situations is generally an ill-posed problem, they removed the ill-posedness by imposing appropriate assumptions. This assumption decouples the probability of observation of the $(i,j)$ element of the matrix into an index depending part $t_{ij}$ and a non-negative function $s(Y_{ij})$ of the matrix value $Y_{ij}$, that is, $Pr(i,j)=t_{ij} s(Y_{ij})$, which makes $s$ and $t$ not appear explicitly in the conditional likelihood.

The paper carefully discusses the difficulties of scale equivalence and shift equivalence that arise when formulating the optimization problem naively under the assumption. The authors also obtain an upper bound on the error of the missing estimation in the Frobenius norm under reasonable assumptions. According to the author's description in L19, methods for estimating missingness in MNAR with such theoretical guarantees are very limited. Finally, they experimentally verified the usefulness of the proposed method on real and synthetic data.

**Strengths:**

- They seem to have established a nonparametric method to prevent model bias.

- They theoretically derived an upper bound on the error in missing values estimation. Since low-rank matrix completion is a well-known NP-hard problem, I believe their error bound has an important contribution to the machine learning community.

- A general theory is developed since almost no assumptions are imposed on the nonnegative functions $s$ and $t$. For example, their theory can also deal with non-uniform missing settings.

- The missing value estimation problem is formulated as a convex optimization problem. This is a very nice contribution.

- It is elegant that $s$ and $t$ do not appear explicitly in the conditional likelihood due to their appropriate assumption.

**Weaknesses:**

-	The biggest concern is that the source code is not available. For example, the paper does not explain how to implement POCS in Algorithm 1. In this situation, there is still a concern about reproducibility without the source code. According to the checklist, the computations can be performed on a personal computer, yet it is better to describe the experimental environment, preferably identifying the operating system, RAM, and programming language.
-	It is worth highlighting that their contribution is to remove ill-posedness through the appropriate assumption. However, it is necessary to define the term "ill-posed" properly and to prove why assumption 1 makes the missing value estimation a well-posed problem. Please let me know if I miss something at this point.
-	Although the experimental results with real data are not promising, this should not be a reason for rejection, given the theoretical guarantees and other strong contributions.

Citations:
- L166: please cite something to support your claim “Nuclear norm regularization has been commonly used to promote low-rankness in the estimation”.
- L100: please cite something to support your claim “Matrix completion under general MNAR is ill-posed”.

Notation:
- For probability distribution, the authors use multiple symbols such as $P$, $Pr$, $p$, and $\mathbb{P}$. I did not get how the authors make the difference among them.
- Some symbols are not defined clearly.
   - L161: I could not find the definition of the inner product between two matrices.
   - L177: What is the symbol “N”? Also, in L178, the authors said, “phi is the pdf,” but I think phi is the dispersion parameter. Please let me know if I misunderstood something.
   - L199: It is better to state clearly that $[n]$ is the set of natural numbers less than or equal to a natural number $n$.

Minor comments:
- What is the definition of the non-negative function in L106? $s:\mathbb{R}\rightarrow\mathbb{R}^+$? If so, please write it clearly.
- The computational complexity of the proposed method should be described.
- L82: In my understanding, $Y_{ij}$ is assumed to be sampled from $f_{ij}(y \mid A, \phi)$. So, if they write $Y_{ij} \sim f_{ij}(y \mid A, \phi) = \dots $ in Equation (1), it is very kind to readers.
- L196: It would be kind to note again that $m_1$ and $m_2$ are the sizes of the matrix A.
- L656: $T_{i,j}$ → $T_{ij}$, $Y_{i,j}$ → $Y_{ij}$

**Questions:**

- L88: I would like to confirm the meaning of “independent” here. You mean $Y_{ij}$ and $T_{ij}$ are not independent of each other, but the tuple $(Y_{ij}, T_{ij})$ is independent of $(Y_{kl}, T_{kl})$ for all $k$ and $l$. Is that right?

- In Appendix B, why do they use the symbol $a(y)$ instead of $s(y)$?

[Edit: after rebuttal]
As the authors addressed my comments appropriately, I raised my score from 6 to 7.

**Limitations:**

The authors adequately addressed the limitations related to shift equivalence and scale equivalence in Section 3.

---

> ### Author Rebuttal · Authors · 2024-08-06
>
> Thank you for the detailed review of our paper.
> ## Weakness
>
> **W1: Please see our global rebuttal.**
>
> **W2: It is worth highlighting that their contribution is to remove ill-posedness through the appropriate assumption. However, it is necessary to define the term "ill-posed" properly and to prove why assumption 1 makes the missing value estimation a well-posed problem. Please let me know if I miss something at this point.**
>
> **A:** Here, ill-posedness refers to the non-identifability issue.The informative missing usually introduces an ill-posed problem not only in matrix completion but also in typical regression (see [14] and [20] cited in the paper). We would like to clarify that we didn’t claim the ill-posedness is completely removed. With assumption 2.1, the problem still possesses some identifiability issues as we mentioned in Section 4 (constant shift and scaling). We show that the proposed pseudolikelihood can recover the remaining identifiable equivalence class  (at a near-optimal asymptotic convergence rate of the standard well-posed (non-informative missing) setting). The problem is still ill-posed, however with pseudolikelihood, we can recover some useful information from it. A contribution of our work is to provide a solid understanding of what is identifiable and can be recoverable reliably.
>
>
> **W3: Although the experimental results with real data are not promising, this should not be a reason for rejection, given the theoretical guarantees and other strong contributions.**
>
> **A:** Thanks for valuing our contributions. For the experiment results, we would like to add some additional insight. Our method targets the challenging MNAR setup where many existing methods fail. But, there is no free lunch. To avoid the bias caused by the MNAR, the proposed method in principle avoids using certain information in the data. Although it alleviates bias under complex missing mechanisms, one more crucial impact that we showed in our work is that the proposed method cannot identify shift-scale equivalence. In general, it is expected that the proposed estimator is less efficient compared to some existing methods under MAR or uniform missingness. This efficiency loss is a price that we pay for the robustness against the violation of restrictive missingness assumptions like MAR and uniform missingness. In other words, we expect our method to behave like the following. For uniform or MAR missingness, our method would be less efficient but remains competitive. For (strong/moderate degree of) MNAR under pairwise missingness, our method would perform better than many existing methods. More importantly, many existing methods would be inconsistent in general. In the experiments, our method performs robustly well for all three datasets, while a few alternatives perform very well in one, but very badly in another dataset. This aligns with the above explanation.
>
> **W4: Citations**
>
> **A:** Thanks for pointing it out. We will explicitly cite related works [23, 24] in the paper to support the claim “the nuclear norm regularization has been commonly used to promote low-rankness” and add a review paper [Hu] to it. General MNAR is ill-posed not only in matrix completion but also in regression [27, 30] and statistical inference [25] in general. A review paper [Tang] will be added too.
>
> **W5: Notation**
>
> **A:** We are sorry for the notation errors in the paper.
> Yes P, Pr, p and $\mathbb{P}$ are all for probability distribution. We will unify the notation to Pr in the paper.
> We will add the definition of the inner product between two matrices “$\langle A, B  \rangle = \sum_{i,j} A_{i,j} B_{i,j}$ for any matrices $A,B \in \mathbb{R}^{n_1\times n_2}$”  in the beginning of Page 5.
> We use N to denote the normal distribution. We will change it to $\mathcal{N}$ to make it more clear. Also, we will replace the notation $\phi$ with $p_{\mathcal{N}}$ for p.d.f. of standard normal distribution, to avoid notation abuse.
> We will add the definition of [n] as you mentioned in the beginning of Section 5.
>
> **W6: Minor comments**
>
> **A:** Thank you so much for the careful checking. We will make corresponding changes.  We will add formal definition of nonnegative function: $s: \mathbb{R} \rightarrow \mathbb{R}^+$ in L106. We will add $Y_{ij} \sim f_{ij}(y\mid A,\phi)$ in L82. We will remind readers that $m_1$ and $m_2$ are the dimensions of matrix $A$ in the beginning of Section 5. Also, we will change the notation $T_{i,j}$ and $Y_{i,j}$ to $T_{ij}$ and $Y_{ij}$ respectively.
>
> ## Questions
>
> **L88 and Appendix B**
>
> **A:** Yes. The independent means $(Y_{i,j}, T_{i,j})$ and $(Y_{k,l}, T_{k,l})$ are independent of each other for $i\neq k$ or $j\neq l$. We will modify $a(y)$ to $s(y)$ in Appendix B to unify the notation.
>
> ## References
>
> [Hu] Zhanxuan H., Feiping N., Rong W., and Xuelong L. (2021). Low rank regularization: A review. Neural Networks.
>
> [Tang] Niansheng T., and Yuanyuan J. (2018). Statistical inference for nonignorable missing-data problems: a selective review. Statistical Theory and Related Fields.

---

> > ### Comment · Reviewer_CJYv · 2024-08-09
> >
> > Dear authors,
> >
> > I appreciate your informative responses. It makes sense that the problem is still ill-posed. As I understand it, a problem is ill-posed when either (i) there is no solution, (ii) the solution is not unique, or (iii) the solution is not continuous to the initial value. However, the usage of the term varies from community to community. Although ill-posedness seems to imply non-identifability in statistics, I would encourage the authors to write more clearly that ill-posed means non-identifability in the main text.
> >
> > Now, the source code is submitted, and the provided insights for the experiments are reasonable. I am totally satisfied with your response. The paper is well-written and will be more readable after you will fix some notations. Thus, I raise my score from 6 to 7.
> >
> > Thanks again for your kind responses and for considering fixing the issue of notations.

---

> ### Author Response · Authors · 2024-08-09
>
> Dear Reviewer CJYv,
>
> Thanks for your appreciation! We will be sure to clarify ill-posedness and clean up all the notations in the revision.
>
> Best,
>
> Authors of Submission 14957

---

### Official Review · Reviewer_Pbx3 · 2024-07-09

**Soundness:** 3
**Presentation:** 3
**Contribution:** 3
**Rating:** 7
**Confidence:** 5

**Summary:**

The paper proposes a novel matrix completion method where the entries are missing not at random. The author(s) tackle the problem using a penalized pairwise pseudo-likelihood approach. The key contributions of the paper are (a) a flexible separable observation probability assumption that depend on the measurements and (b) to provide a novel asymptotic result concerning the concentration bound on the proposed estimator. The success of the proposed method is validated via numerical experiments and on three real data analysis.

**Strengths:**

The main contribution highlighted in the work is how to perform matrix completion when the entries are not missing at random (MNAR). The fundamental assumption on the observation probability given the measurements is based on a separability assumption which tacitly helps avoiding the non-identifiability issue typically faced in MNAR setting.

Some other strengths of the paper are the following-

(a) novel use of pairwise pseudo-likelihood approach to solve the missing completion problem in MNAR setting

(b) novel theoretical result concerning the concentration bound on the proposed matrix estimator

(c) well written clarification (with appropriate examples) of the non-identifiability issue in the MNAR setting

**Weaknesses:**

Following are some of the weaknesses of the paper

(a)  A bit more discussion on how the constants $\kappa$ and $\rho$ influence the right hand bound on equation (7) would be insightful.

(b) Some comparison of the computational times of the competing approaches vs the proposed approach would be helpful.

(c) In the simulations, apart from Gaussian distribution mechanism, it would have been interesting to see situations where observations have been generated from a skewed distribution.

**Questions:**

I have the following queries for the authors-

(a) In assumption 2.1 is it required to have $s(Y_{ij})$ distinct at least for one pair $(i,j)$ and $(i^{'},j{'})$ ? Otherwise we may end up with a situation where the assumption 2.1 is another variant of non-uniform missing structure scenario.

(b) Why do we need the constraint $\ || A || \_{\infty} \leq a $ in the optimization described in (6) ?

(c) Computational cost vs performance gain should be examined in both real data and simulations to understand the efficacy of the proposed method.

(d) One thing that comes out in Table 1 tobacco dataset is that the standard error of pairwise pseudo-likelihood is the second highest among all the methods. It may be interesting to check through the simulations if the standard error of pairwise pseudo-likelihood  is in general remain high?

**Limitations:**

The authors have adequately addressed the limitations of the work.

---

> ### Author Rebuttal · Authors · 2024-08-06
>
> We would like to thank you for the constructive feedback and review.
>
> **Weakness: A bit more discussion on how the constants $\kappa$ and $\rho$ influence the right hand bound on equation (7) would be insightful.**
>
> **A:**
> Thanks for the suggestion. When $\kappa$ is small, it means there exists one pair of indices such that $E((Y_{ij}-Y_{i’j’})^2)$ (other parts of $E [Z^2_{i,j,i’,j’}]$ is positive) is close to 0, and it falls into the difficult case to identify either of them. For $\rho \in [1, \infty)$, when $\rho=1$, it refers to the uniform missing setting, which is the simplest setting to deal with.  When $\rho$ gets larger, there is more variability among $\pi_{i,j}$ and so the bound in (7) is larger.
>
> **Q1: In assumption 2.1 is it required to have $s(Y_{ij})$ distinct at least for one pair $(i,j)$ and $(i’,j’)$? Otherwise we may end up with a situation where the assumption 2.1 is another variant of non-uniform missing structure scenario.**
>
> **A:** It’s correct that if $s(Y_{ij})$ are all the same, the missing mechanism is degenerated to non-uniform missing. But we don’t require that assumption that at least a pair of $s(Y_{ij})$ are distinct. Here the implication is that our method also applies to non-uniform missing scenarios. The fact that our method works for a wide range of scenarios is practically appealing.
>
> **Q2: Why do we need the constraint $|A|_\infty \leq a$  in the optimization described in (6)?**
>
> **A:** This constraint was used in our theoretical analysis, and is common in matrix completion literature such as [5] and [32] cited in the paper. Practically, the effect of this constraint is mild or non-existent, as it is common to pick a very large $a$. Indeed, some prior works might drop this constraint completely in the practical algorithm, which typically speeds up the optimization.
>
> **Q3: Please see our global rebuttal for question/weakness on numerical experiments.**
>
> **Q4: One thing that comes out in Table 1 tobacco dataset is that the standard error of pairwise pseudo-likelihood is the second highest among all the methods. It may be interesting to check through the simulations if the standard error of pairwise pseudo-likelihood is in general remain high?**
>
> **A:** That is a good point. We actually included standard errors in simulations. There are error bars in Figure 2, but they are small and hard to recognize. We will make it more obvious in the revision. We also had a table in the Appendix showing the standard errors (Table 2 in Appendix C). The standard error of pairwise pseudolikelihood is small. The performance issue in Tobacco dataset might be because the assumption of the proposed method is not satisfied  (if one entry is missed, the subsequent entries are also missed)

---

> > ### Comment · Reviewer_Pbx3 · 2024-08-11
> > **Comments to the rebuttal**
> >
> > I am happy with the authors' detailed responses to my queries. Most of the theoretical issues have been clarified and the authors provided a detailed numerical comparison with the existing methods in terms of computational times.
> >
> > I think they have done a great job in answering in detail all the other queries raised by the reviewers. I am happy to keep the score at 7 and would support the "acceptance" of the paper.

---

> > > ### Author Response · Authors · 2024-08-11
> > >
> > > Thank you for your feedback and support! We appreciate your acknowledgment of our efforts in rebuttal. Your constructive feedback is invaluable in refining our work.

---

### Official Review · Reviewer_1V66 · 2024-07-10

**Soundness:** 3
**Presentation:** 3
**Contribution:** 3
**Rating:** 7
**Confidence:** 2

**Summary:**

Current matrix completion approaches mostly deal with non-uniform sampling scenarios. However, few  approaches can allow the missingness to depend on the mostly unobserved matrix measurements. In this paper, the authors propose a regularized pairwise pseudo-likelihood approach for matrix completion in this scenario. The proposed estimator can recover the low-rank parameter matrix asymptotically up to an identifiable equivalence class of a constant shift and scaling, at a near-optimal asymptotic convergence rate of the standard well-posed setting, while effectively mitigating the impact of informative missingness.

**Strengths:**

This paper considers the matrix completion problem under the informative missing scenario and applies the pseudo-likelihood method to matrix completion for the first time, which is relatively novel. This approach can effectively mitigate the impact of informative missingness with some theoretical guarantees.

**Weaknesses:**

However, there is a lack of literature review related to the informative missing setting and a lack of comparison with related methods aimed at the informative missing setting, making it difficult to accurately evaluate the contribution.

Also, I am unsure if these scenarios are widely applicable. In the experimental section, the proposed method does not achieve the best results on some datasets. For instance, on the Tabacco Dataset, the method does not outperform the SNN method.

Lastly, in line 350, the author says the proposed method's grouping nature adds an extra computational burden, thus this approach seems not to be computationally efficient.

**Questions:**

1)	I still am unclear about the meaning of “informative missingness”, could the author give a more detailed definition and a concrete example?

2)  Are the authors the first to consider this informative missingness setting in matrix completion? And are there any other methods to address informative missingness aimed at matrix completion? And How to evaluate the strengths and weaknesses of these methods ( including the method proposed in this paper).

3)	What is the difference between the pseudo-likelihood method and Bayes modeling?

4)	Please provide a comparison of the computational time between the proposed method and the previous algorithms for matrix completion tasks. This is crucial as people often deal with high-dimensional and massive datasets, where computational efficiency is a key factor. If possible, please also provide the errors (TRMSE) versus the iterations/time during the trajectories.


5)	It might be challenging to effectively identify whether a scenario qualifies as "informative missingness" and apply this approach. For instance, on the Tabacco Dataset, the method does not outperform the SNN method. Is this result arise from this phenomenon/dataset that doesn’t belong to the “informative missingness” scenario? How to identify a phenomenon/dataset belonging to the “informative missingness” scenario?

**Limitations:**

1) I'm uncertain about the article's claim that "informative missingness" is widespread and significant in real-world situations. It might be challenging to effectively identify whether a scenario qualifies as "informative missingness" and apply this approach. For instance, on the Tabacco Dataset, the method does not outperform the SNN method. This dataset may not fall into the “informative missingness” scenario. How to identify whether a phenomenon is "informative missingness"?

2) There is not an extensive literature review on informative missingness in matrix completion or other related problems, please provide a detailed literature review about informative missingness and related approach comparisons.

3)  In line 350, the author says the proposed method's grouping nature adds an extra computational burden and I am not sure whether this approach is computationally efficient in practice.

---

> ### Author Rebuttal · Authors · 2024-08-05
>
> We greatly appreciate the effort you have spent reviewing our paper.
>
> **Q1: Could the author give a more detailed definition of informative missingness and a concrete example?**
>
> **A:** We used “informative missingness” to refer to what statisticians often called Missing Not At Random (MNAR) assumption [LR], in which the probability of missingness can depend on the value of the variable that is not observed.  In ratings of movies, for example, people are more likely to watch movies that they may like due to unknown preferences and those preferences would also affect the rating, the missing probability is therefore likely dependent on ratings which may be unobserved.
>
> **Q2: Are the authors the first to consider this informative missingness setting in matrix completion? And are there any other methods to address informative missingness aimed at matrix completion? And How to evaluate the strengths and weaknesses of these methods?**
>
> **A:** Please see our global rebuttal about related works and theoretical evaluation. As for evaluating the numerical performance of these methods, we require independent sampling of validation data or known generating mechanisms. In the simulation study, we generate data under the condition satisfied for our setting, and we have the validation data to evaluate the completion errors. Our methods outperform other methods. In the Tabacco real data, we create the missing mechanism following the steps described in [2], which satisfies the conditions in [2] but violates the assumption in our setting, and our method still gives robust performance. Please see more details in the global rebuttal as well.
>
> **Q3: What is the difference between the pseudo-likelihood method and Bayes modeling?**
>
> **A:** The pseudo-likelihood is not a full likelihood that is used in conventional Bayesian modeling.  Indeed, Bayesian modeling for the missing not at random settings will require parametric modeling of the full distribution of the outcomes and missingness mechanism, which is usually viewed as a daunting task.  Because of identifiability issues, the result from Bayesian modeling will typically depend heavily on the prior used, which is another undesirable feature. The pseudo-likelihood used eliminates unknown $t_{ij}$ and function $s(\cdot)$, which allow us to focus on the estimation of the target matrix.
>
> **Q4: Please see our global rebuttal.**
>
> **Q5: It might be challenging to effectively identify whether a scenario qualifies as "informative missingness" and apply this approach. For instance, on the Tabacco Dataset, the method does not outperform the SNN method. Is this result arise from this phenomenon/dataset that doesn’t belong to the “informative missingness” scenario? How to identify a phenomenon/dataset belonging to the “informative missingness” scenario?**
>
> **A:**  Yes, you are right that it is challenging to effectively identify informative missingness / missing not at random (MNAR). Indeed, it is known in the missing data literature that available data is generally unable to distinguish between missing at random and among MNAR models [4, 5]. As in page 4 of [D], "It is impossible to test whether the MAR condition is satisfied, and the reason should be intuitively clear. Because we do not know the values of the missing data, we cannot compare the values of those with and without missing data to see if they differ systematically on that variable." Therefore, a fundamental limitation in any missing data analysis is that the missing mechanism has to be somewhat assumed. Given this fundamental challenge, our goal here is to adopt a flexible enough missing mechanism so that the methods still work for a wide range of scenarios to alleviate the concerns of mis-specifying missing mechanisms.
>
> The separable assumption that we assumed indeed includes uniform missing, missing-at-random and the focused MNAR assumption [BH] as special cases, and has been argued as a general but workable assumption [34]. For matrix completion, the separable assumption that we use in this work is already fairly relaxed compared to those in the missing data literature, since each entry could have a complicated observation probability based on its location as well as a function on the potential observation (even though it could be missing). In terms of location dependence, it is allowed to be completely flexible. Among matrix completion methods (especially those with theoretical guarantees), our assumption covers the uniform missing and missing-at-random settings, which is the focus of most matrix completion literature. In other words, in terms of the applicability of the missingness assumption, our work is generally more applicable than many existing literature that assumes uniform missing and missing-at-random settings. When those settings fail, our assumption could still hold and our method will still have theoretical guarantee while others may fail.
>
> As for the Tabacco dataset, we note that the way to generate missingness is adapted from the SNN paper. SNN is well-designed for the underlying missing mechanism. When one entry is missed in Tobacco dataset, the entries in the following period are also missed. This does not satisfy the assumption of our work. So it is not surprising to see our method perform sub-optimality. However, the performance of our method still remains strong. This example can be used to illustrate a situation where the assumption of our method fails.
>
> **References:**
>
> [LR] Little R J A, Rubin D B. Missing data in large data sets. Statistical Methods and the Improvement of Data Quality. Academic Press, 1983: 215-243.
>
> [D] Allison, P.D. (2002) Missing Data, Quantitative Applications in the Social Sciences, Thousand Oaks, CA:SAGE.
>
> [LR2] Little R. J. and Rubin D. B. (2019) Statistical analysis with missing data, Hoboken, NJ:Wiley.
>
> [BH] Gomer B. and Yuan K.H. (2021) Subtypes of the missing not at random missing data mechanism. Psychological Methods, 26(5), 559-598.

---

> > ### Comment · Reviewer_1V66 · 2024-08-10
> >
> > I am grateful for the meticulous reply from the author. Consequently, I have decided to upgrade my rating from 6 to 7

---

> > > ### Author Response · Authors · 2024-08-10
> > >
> > > We are glad we could provide the clarity you needed. Thank you again for your comments and feedback, which helped improve the quality of our work!

---

### Author Rebuttal · Authors · 2024-08-06

We’d like to thank the efforts of all reviewers. The reviews are insightful and constructive to improve our work. We address some common questions in this global rebuttal and leave other questions in the detailed response to each reviewer.

We first discuss **the related works and evaluation of strength and weaknesses of them**. To the best of our knowledge, we are one of the first several works that consider the missing not at random setting in matrix completion and have solid theoretical guarantees. [2] claims that they can deal with MNAR setting. However, they assume selections and noise are independent conditioned on latent factors, as shown in their Assumption 2. On the contrary, our setting allows missingness to depend on noise. [HYF] also deals with informative missing in matrix completion. However, they need additional covariate Information to complete the matrix.
To evaluate the theoretical strengths and weaknesses of these methods, one way is to compare the assumptions for the proposed methods. Compared to the above two works, our setting is more general as we do not require the independence between selections and noise given the true matrix and we do not need additional covariate information. However, we do require that $T_{i,j}$ is independent of $T_{k,l}$ for $i\neq j$ or $k\neq l$, while [2] allows selection to be dependent among different entries. Another way to evaluate is to compare the theoretical bound for the proposed estimators. [2] need additional technical conditions to develop finite sample error bound, and their bound is point-wise, i.e., the bound is for a given location $||\hat A_{i,j} - A_{i,j}||$. [HYF] also needs additional conditions for the likelihood and restricted eigenvalues in order to obtain the convergence. Our error bound is developed under relatively weak conditions and achieves the minimax convergence rate.

Regarding the concerns of the **computational time**, we conducted a **computational complexity** analysis of our method. Note that the computational complexity of the pairwise pseudolikelihood is $O(n^2)$, where $n=|\mathcal{E}|$, the number of observations. The gradient of it is therefore $O(n^2)$ from Eq. (8). The soft thresholding operator relies on the SVD with a possibly non-low rank matrix, which takes $O(m_1 m_2 m)$ Flops. POCS (projection onto convex sets) takes $O(m_1 m_2)$. Combining them all, our algorithm takes $O(n^2 + m_1 m_2 m)$ Flops at each iteration. When the matrix has less observed entries, the second term dominates and our algorithm is as efficient as other SVD based methods. However, when the matrix has relatively more observed entries, the first term dominates, which could scale as $O(m_1^2 m_2^2)$ in the worst case.


We added **comparisons of the computational time** regarding the setting in Figure 2 ($\sigma^2=1$). The computational times are listed as follows. While incorporating more complex missing mechanisms, our method and SNN also take the most time. One practical way to speed up the computation of our method is to use a stochastic version of Algorithm 1 (i.e., training in batches). The focus of this paper is more on the robust recovery when encountering informative missing, and less on the computational efficiency with the knowledge that it could be theoretically slower than other SVD based methods. But our method is still faster than SNN, where both methods consider more complex missing mechanisms. Given the promising statistical properties of the proposed method, a future direction is to develop scalable algorithms for the proposed estimator or its variants.

|    | Time (s) |
| -------- | ------- |
| SoftImpute  |  0.01$\pm$0.005   |
| CZ | 0.22$\pm$0.09     |
| MFW    |  1.90$\pm$0.75  |
|SNN| 15.23$\pm$1.39|
|Ours| 8.67$\pm$1.23|

We’d like to add more details about our **experimental environment**. The experiments were run on a MacBook Pro, 2021 with Apple M1 Pro chip and 16GB RAM. The POCS (projection onto convex sets) in our algorithm is the projection onto the intersection of the nuclear norm ball and centered hyperplane. Both projection can be achieved quite effectively via truncating large values and eliminating the mean of all the entries. The POCS here refers to running these two projections iteratively until convergence. We will clearly describe these steps in the revision.


We originally planned to make the source code public on GitHub after the review period. We'd like to provide an anonymous repo about the implementation details for review purpose now. The rebuttal policy prevents us from sharing a link here. We have shared it with AC via a separate comment and you may request it upon your needs.

## References

[HYF] Jin H, Ma Y, Jiang F. Matrix completion with covariate information and informative missingness. Journal of Machine Learning Research, 2022, 23(180): 1-62.

---

### Public Comment · ~Yuanhong_A1 · 2025-04-04
**Technical Questions on Code and Lemma A.6 Proof**

Thank you for your interesting work! I have some technical questions regarding the implementation and theoretical analysis that I would appreciate your clarification on:

**Code Implementation**

I followed your code in https://github.com/jiangyuan-li/mc-w-pseudolikelihood, and found something that confused me.

1. **Gradient Accumulation in Pseudo-Solver**
In `src/trainers.py` (pseudo_solve function), the current implementation calls `loss.backward()` without first zeroing gradients (e.g., via `model.zero_grad()`). During my reproduction attempts, I observed the performance in the pseudo-likelihood estimator wasn't as good as expected, which may due to unintended gradient accumulation across optimization steps. Could you confirm whether this was an implementation oversight?

2. **Metric Consistency Across Methods**
In `src/utils.py`, the evaluation metric for the proposed method (`get_error_w_shift_scaling`) employs regression-based error correction before RMSE calculation, while baseline methods use direct RMSE (`get_error`). Could you discuss whether this methodological discrepancy might affect the fairness of comparisons? If intentional, would it be possible to provide ablation studies showing the impact of this design choice?

**Theoretical Analysis**

Regarding Lemma A.6:
The definition of $\\xi_{n,g}$ in Eq.11 (Page15) involves a maximum on $i,j$, yet the subsequent summation $\\sum_{g=1}^{n-1} \xi_{n,g}$ appears to exchange the order of maximization and summation. Specifically, the current derivation claims:

$$
\sum_{g=1}^{n-1} \xi_{n,g} = \max\left\{ \sum_{i=1}^{m_1} \sum_{j\neq j'}^{m_2} T_{ij}T_{ij'},\  \sum_{j=1}^{m_2} \sum_{i\neq i'}^{m_1} T_{ij}T_{i'j} \right\},
$$
at the bottom of Page 22.

However, it seems the left side represents a sum of maxima, while the right side is a maximum of sums, implying the original statement should instead be an inequality (LHS $\\geq$ RHS). This appears to create a gap in Lemma A.6's proof, and consequently in Lemma A.1's dependencies.

Have you considered alternative approaches to bound this term while preserving the proof structure? I would be grateful for your insights on resolving this technical challenge.

---

> ### Public Comment · ~Jiayi_Wang7 · 2025-04-04
> **Response to your question about theoretical analysis**
>
> Dear Yuanhong,
> Thanks for carefully checking our proof. Really appreciate! I will help to clarify the theoretical analysis.
> Yes the change of maximum and summation is a bit problematic. Indeed, instead of taking maximum in the definition of \xi_{n,g}, we can take the direct sum of these two components, which leads to bounding the summation of two summations in Lemma A.6. While the order of these two summations is the same, it will eventually lead to the same order bound.

---

> ### Public Comment · ~Jiangyuan_Li1 · 2025-04-04
> **Responses**
>
> Thanks Yuanhong. For Q1, we had a gradient zeroing out step "model.mu.weight.grad.data.zero_()" after each gradient descent step, it was deleted wrongly when we cleaned up the code for submission (while deleting the comments used for our own purposes). Please include it in your implementation. Hope that doesn't bother you too much. For Q2, we made it clear that our method suffers from constant shift and scaling when reporting error metrics like RMSE and MAE. Therefore, we did a simple linear regression on validation dataset to mitigate such bias. Other methods also rely on validation dataset for hyper-parameter tuning. You may view the constant shift and scale as hyperparameters of our method (in some sense), which are tuned on validation data. Thanks again for the careful check and valuable comments. Hope you enjoyed reading our work!

---

### Decision · Program_Chairs · 2024-09-25

**Decision:**

Accept (spotlight)

**Comment:**

Good paper.